**1** **Different responses of ecosystem $CO_2$ and $N_2O$ emissions and CH4 uptake to**

**2** **seasonally asymmetric warming in an alpine grassland of the Tianshan**

**3** **Mountains**

**4**    Yanming Gong[1], Ping Yue[2], Kaihui, Li[1], Anwar Mohammat[1]*, Yanyan Liu[1]*

**5** [1]State Key Laboratory of Desert and Oasis Ecology, Xinjiang Institute of Ecology and

**6**    Geography, CAS, Urumqi, 830011, China

**7** [2]Urat Desert-Grassland Research Station, Northwest Institute of Eco-Environment and

**8**    Resources, CAS, Lanzhou, 730000, China

**9** *Correspondence: Anwar Mohammat and Yanyan Liu, Email: liuyany@ms.xjb.ac.cn

**10** **Abstract:**

**11**    An experiment was conducted to investigate the effect of seasonally asymmetric

**12** warming on ecosystem respiration ($Re$), CH4 uptake, and $N_2O$ emissions in alpine

**13** grassland of the Tianshan Mountains of Central Asia, from October 2016 to

**14** September 2019. The annual mean of $Re$, CH4, and $N_2O$ fluxes in growing season

**15** were 42.83 mg C m$^{-2}$ h$^{-1}$, −41.57 μg C m$^{-2}$ h$^{-1}$, and 4.98 μg N m$^{-2}$ h$^{-1}$, respectively.

**16** Furthermore, warming during the non-growing season increased $Re$ and CH4 uptake

**17** by 7.9% and 10.6% in growing season, 10.5% and 9.2% in non-growing seasons,

**18** respectively. However, the increase in $N_2O$ emission in the growing season was

**19** mainly caused by the warming during the growing season (by 29.7%). the warming

**20** throughout the year and warming during the non-growing season increased $N_2O$

**21** emissions by 101.9% and 192.3% in non-growing seasons, respectively. The $Re$, CH4

**22** uptake, and $N_2O$ emissions were positively correlated with soil temperature. Our

results suggested that $Re$, $CH_4$ uptake and $N_2O$ emissions were regulated by soil

temperature, rather than soil moisture, in the case of seasonally asymmetric warming.

In addition, the response rate was defined by the changes in greenhouse gas fluxes

driven by warming. In our field experiment, we observed the stimulatory effect of

warming during the non-growing season on $Re$ and $CH_4$ uptake. In contrast, the

response rates of $Re$ and $N_2O$ emissions were gradually attenuated by long-term

annual warming and the response rate of $Re$ was also weakened by warming over the

growing season. These findings highlight the importance of warming in the

non-growing season in regulating greenhouse gas fluxes, a finding which is crucial for

improving our understanding of C and N cycles under the scenarios of global

warming.

**Keywords:** Alpine steppe; Extreme climatic event; Greenhouse gas fluxes;

Warming of open-top chambers

**1. Introduction**

Since the industrial revolution, human activities have intensified global warming.

The global surface temperature increased by about 0.85°C from 1880 to 2012 (IPCC,

2013). Furthermore, the temperature is expected that the surface temperature will

increase by about 1.1–6.4°C by the end of this century (IPCC, 2007, 2013). The rise

in atmospheric temperature over the year is not continuous on the temporal scale but

there is asymmetrical warming across the seasons (Xia et al., 2014). The 3rd and 4th

Assessment Report of the Inter-Governmental Panel on Climate Change (IPCC)

proposed that, against the backdrop of global warming, the temperature change shows

that the warming amplitude in the winter is greater than that in the summer, with the
warming amplitude at high latitude being greater than that at low latitude, and
confirmed that the warming shows asymmetric trends on a seasonal scale (Easterling
et al., 1997; IPCC, 2001, 2007).

49         Carbon dioxide ($CO_2$), methane ($CH_4$), and nitrous oxide ($N_2O$) are three of the

major greenhouse gases (GHGs) in the atmosphere that directly cause global climate
warming, with their contributions to global warming being 60%, 20%, and 6%,
respectively (IPCC, 2007, 2013). Experimental warming is known to influence
ecosystem respiration ($Re$), $CH_4$ uptake, and $N_2O$ emission (Pärn et al., 2018; Treat et
al., 2018; Wang et al., 2019). Information on $Re$, $CH_4$ uptake, and $N_2O$ emission will
enhance our understanding of ecosystem C and N cycling processes and improve our
predictions of the response of ecosystems to global climate change (Li et al., 2020;
Wang et al., 2019).

58         At present, most studies focus on the influence of warming on GHG flux in

terrestrial ecosystems during the summer months (Keenan et al., 2014; Li et al., 2011;
Yang et al., 2014). Nevertheless, data on the influence of asymmetric warming on the
GHG flux on a seasonal scale are scarce. A study of the Alaskan tundra found that
summer warming (using open-top chambers to increase air temperatures in the
growing season) significantly increased $Re$ in the growing season by about 20%
(Natali et al., 2011). Compared with the slight effect of winter warming on the
ecosystem respiration in the growing season, warming increased ecosystem
respiration during the snow-covered non-growing season by more than 50% (Natali et
al., 2011). Lin et al. (2015) reported that the response of soil $CH_4$ uptake rates to
temperature increases in alpine meadows of the Qinghai-Tibet Plateau were not
consistent seasonally, with $CH_4$ uptake in the non-growing season being more
sensitive to temperature (increasing by 162%) than the corresponding value in the
growing season. A study by Cantarel et al. (2012) in an alpine grassland ecosystem
showed that the response of $N_2O$ emission to warming showed clear seasonal
differences, with the $N_2O$ emission in the growing season showing significant
differences between the warming treatments, whereas the response of $N_2O$ emission
to the warming treatments in November was not obvious. A recent study showed that
seasonal variations in carbon flux were more closely related to air temperature in the
meadow steppe (Zhao et al., 2019). Another study found that experimental warming
enhanced $CH_4$ uptake in the relatively arid alpine steppe, but had no significant effects
on $CH_4$ emission in the moist swamp meadow (Li et al., 2020). Furthermore, soil $CH_4$
uptake was not significantly affected by warming in the alpine meadow of the Tibetan
Plateau (Wu et al., 2020). In contrast, a global meta-analysis showed that
experimental warming stimulates ecosystem respiration in grassland ecosystems, and
the response of ecosystem respiration to warming strongly varies across the different
grassland types, with greater warming responses in cold than in temperate and semi-
arid grasslands (Wang et al., 2019). Across the data set, Li et al. (2020) demonstrated
that $N_2O$ emissions were significantly enhanced by whole-year warming treatments.
In contrast, no significant effects on soil $N_2O$ emissions were observed by in
short-season warming.
In summary, the GHG fluxes in terrestrial ecosystems shows significant
interannual, and seasonal variations, and its response to warming also varies over
different temporal scales. After long-term uniform warming, the biotic and abiotic
factors have adapted to the temperature increase, and the GHG fluxes response to
increasing temperature is smaller than that in the early stages of warming. For
example, over longer time periods of warming, accelerated carbon decomposition and
increased plant N uptake may decrease soil organic C and N pools (Wu et al., 2012),
and the microbial community with variable C use efficiency may reduce the
temperature sensitivity of heterotrophic respiration (Zhou et al., 2012). Moreover,
climate warming is often unstable, with most of it occurring as extreme events
(Jentsch et al., 2007). The heterogeneity of warming may change the adaptability of
GHG fluxes to warming, and thus affect the carbon and nitrogen cycles in terrestrial
ecosystems. In this study, we hypothesize the stimulatory effect of warming during the
non-growing season on $Re$, $CH_4$ uptake and $N_2O$ emissions. However, the response
rates of $Re$, $CH_4$ uptake and $N_2O$ emissions were gradually attenuated by long-term
annual warming and warming over the growing season, respectively.
**2.  Materials and methods**
The experiment was conducted from October 2016 to September 2019 at the
Bayinbuluk Grassland Ecosystem Research Station, Chinese Academy of Sciences
(42°52.76′ ~ 42°53.17′ N, 83°41.90′ ~ 83°43.12′ E, 2460 m above sea level), which is
located in the southern Tianshan mountains of Central Asia, Xinjiang Uyghur
Autonomous Region of China. Permafrost is present in the Bayinbuluk alpine

grassland, with the average maximum frozen depth (from 2000 to 2011, Zhang et al.,

2018) being more than 250 cm. The mean annual temperature was −4.8 °C per decade,

with the lowest monthly temperature in January (−27.4 °C) and the highest in July

(11.2 °C), and the mean annual precipitation amounted to 265.7 mm, with 78.1%

occurring during the growing season, from June to September (Geng et al., 2019).

Variations in soil temperature, soil moisture, air temperature and precipitation are

shown in Figure S1, S2, S3 and S4, respectively. The site was fenced since 2005, all

the plots were dominated by *Stipa purpurea*, *Festuca ovina*, *Oxytropis glabra*, and

*Potentilla multifida*. The soil was sub-alpine steppe soil, the parent material of the soil

was Loess, and the average annual soil moisture was 5.9% (2017-2019).

The open-top chambers (OTCs) were made of 5 mm thick tempered glass. To

reduce the impact of precipitation and snow, the OTC was constructed with a

hexagonal round table which was 100 cm high, and the diagonals of the bottom and

top were 100 cm and 60 cm, respectively. Four treatments were simulated using OTCs:

warming throughout the year (AW), warming in the non-growing season (October 1 to

May 31 of the next year) only (NGW), warming in the growing season (June 1 to

September 30) only (GW) and no warming (NW). After the warming in the NGW or

GW, the tempered glass was removed and the frame was retained. Three replicate

plots were established for each treatment, each plot measuring 1 m × 1 m, with a 3-m

wide buffer zone between adjacent plots, making a total of 12 plots. Soil temperature

and soil moisture were measured at a frequency of every half an hour by an outdoor

temperature and humidity data recorder (at 10 cm depth; HOBO U23-001; Onset

Computer Corporation, Bourne, USA). The air temperature inside the OTCs is also
recorded at a frequency of every half an hour using HOBO Pro RH/TEMP Data
LOGGERS (hanged in the center of the OTCs, 50cm above the surface; Onset
Computer Corporation, Bourne, USA). Soil temperature and air temperature were
increased about 2.3 $^{o}C$ and 4 $^{o}C$ by the warming treatment, respectively (Figure S1
and S3). Soil moisture was reduced about 5% by the warming treatment (Figure S2).
$Re$, $CH_4$ and $N_2O$ fluxes were measured using static chambers, made of PVC
tubing with diameter 0.25 m and height 0.17 m, with one chamber in each of the 12
plots. Gas samples were taken 0, 10, 20 and 30 minutes after the lid of the static
chamber was sealed in between 12:00 and 14:00 (GMT + 8), collecting once or twice
a week. The rates of ecosystem respiration, $CH_4$ and $N_2O$ fluxes were calculated
based on the change in concentration of $CO_2$, $N_2O$ and $CH_4$ in each chamber over
time by a linear or non-linear equation ($P < 0.05$, $r^2 > 0.95$) (the positive flux values
represent emission, and the negative flux values represent uptake; Liu et al. 2012;
Wang et al. 2013). Concentrations of individual gases in samples were measured using
a gas chromatograph (GC) (Agilent 7890A; Agilent Technologies, Santa Clara, CA,
USA).
Effects of seasonally asymmetric warming on $Re$, $CH_4$ uptake, and $N_2O$
emissions were analyzed by two-way repeated-measures analysis of variance
(ANOVA). One-way ANOVA was used to compare soil temperature, soil moisture
and air temperature differences, respectively. Nonlinear regression analyses
(Exponential Growth, Single, 3 Parameter) was used to identify the relationship

between ecosystem respiration (Re) and soil temperature (at 10-cm depth) from

October 2016 to September 2019. General linear analyses were used to identify

significant linear correlations and regressions between soil temperature, or soil

moisture variation and the responses of $CH_4$ uptake, or $N_2O$ emissions.

Variation-partitioning analysis was used to disentangled the influence of soil

temperature and soil moisture on Re, $CH_4$ uptake and $N_2O$ emission under the four

treatments in the growing season and the non-growing season, respectively. The

natural logarithm of the response ratio (RR) was used to reflect the effects of

seasonally asymmetric warming on alpine grassland GHG fluxes (Hedges et al., 1999).

The RR is the ratio of the mean value of the chosen variable in the warming group

( $\overline{W}_t$ ) to that in the control group (NW; $\overline{W}_c$), and is an index of the effect of

seasonally asymmetric warming on the corresponding variable (Eq. 1). All statistical

analyses were conducted using SPSS (version 20.0) (IBM, Armonk, NY, USA) with

the statistically significant difference threshold set at $P < 0.05$.

$$RR = \ln\left(\frac{\overline{W}_t}{\overline{W}_c}\right) = \ln\left(\overline{W}_t\right) - \ln\left(\overline{W}_c\right)$$

(1)

**3. Results**

Our study showed that the Bayinbuluk alpine grassland exhibited a low Re, was a

net $CH_4$ sink, and a negligible $N_2O$ source. The annual mean values of Re, $CH_4$ uptake,

and $N_2O$ emissions in the growing season were 42.83 mg C $m^{-2}$ $h^{-1}$, 41.57 µg C $m^{-2}$ $h^{-1}$,

and 4.98 µg N $m^{-2}$ $h^{-1}$, respectively, from October 2016 to September 2019. One-way

ANOVA results of Re, $CH_4$ uptake and $N_2O$ emissions among the four warming

treatments were not significant, with the exception that the soil $CH_4$ uptake in the

growing season 2019 under GW treatment was significantly higher than that of the AW and NGW treatments ($P < 0.05$). Compared with the control group (NW), the $Re$ was decreased by 7.5% and 4.0% in the growing season and non-growing season, respectively, under AW and decreased by 2.4% and 8.5% under GW in the growing season and non-growing season, respectively. However, compared with the control group, the $Re$ under NGW increased by 7.9% and 10.5% in the growing season and non-growing season, respectively, averaged over the three years (Figure 2 a).

The AW temperature change induced a 6.4% increase in $CH_4$ uptake in the growing season and a 3.8% decrease in the non-growing season. The GW treatment resulted in 7.1% and 10.2% increases in $CH_4$ uptake in the growing season and non-growing season, respectively. On the contrary, the NGW generated a 10.6% and 9.2% decrease in $CH_4$ uptake in the growing season and non-growing season, respectively (Figure 2 b). The AW and NGW treatments resulted in 5.8% and 2.2% decreases, respectively, in $N_2O$ emission in the growing season, and 101.9% and 192.3% increases, respectively, in $N_2O$ emission in the non-growing season. Compared with the control, NW group, the $N_2O$ emission increased by 29.7% and decreased by 24.4% under GW in the growing season and non-growing season, respectively (Figure 2 c).

The results of two-way repeated measures ANOVA showed significant interannual differences of $Re$ in the growing season ($P < 0.05$, Figure 1 a), whereas the $CH_4$ uptake under the warming treatment exhibited significant differences in the growing season ($P < 0.01$; Figure 1 b), and the interannual $N_2O$ emission showed

significant differences in both the growing season and non-growing season ($P < 0.05$,
Figure 1 c). Therefore, interannual variation was larger than the impact of the
warming treatment (for $Re$ and $N_2O$ emissions, Figure 1), whereas the warming
treatment had a significant impact on $CH_4$ uptake. Under the four warming treatments,
$Re$ exhibited exponential growth, respectively ($P < 0.05$; Figure S5 a). we observed
increasing $CH_4$ uptake with increasing soil temperature ($P < 0.05$; Figure S5 b). On
the other hand, the $N_2O$ emission showed a significantly positive linear correlation
with soil temperature, but only under NGW ($P < 0.05$; Figure S5 c).
The soil moisture was reduced by warming in the alpine grassland (Figure S2).
However, $Re$, $CH_4$ uptake and $N_2O$ emission were not significant linearly correlated
with soil moisture, respectively ($P \geq 0.05$; Figure S6). We disentangled the influence
of soil temperature and soil moisture on $Re$, $CH_4$ uptake, and $N_2O$ emission by
variation-partitioning analysis under the four treatments in the growing season and the
non-growing season (Figure 4). Under the NGW treatment, $Re$, $CH_4$ uptake, and $N_2O$
emission in the non-growing season were more influenced by soil temperature than by
soil moisture. Under the GW treatment, there was the single effect of soil temperature
on $CH_4$ uptake and $N_2O$ emission in the non-growing season. In contrast, there were
the joint effects of soil temperature and moisture on $Re$ in the non-growing season
under the GW treatment. $Re$ in the growing season was influenced more by soil
moisture than soil temperature under the GW treatment. Annual $Re$ under the AW
treatment was influenced by the joint effects of soil temperature and moisture.
**4. Discussion**
Our study found that the response rate of $Re$ to temperature significantly
decreased with the increase in soil temperature ($\triangle ST_{AW}$ and $\triangle ST_{GW}$) under AW and
GW treatments, respectively (Figure 3 a, c; $P < 0.05$). This finding indicated that the
response of $Re$ to soil temperature became less and less sensitive to soil temperature
with warming throughout the year (or the growing season) in the alpine grasslands.
On the contrary, NGW significantly increased the response rate of $Re$ to temperature
change ($\triangle ST_{NGW}$), indicating that warming in the non-growing season amplified the
sensitivity of $Re$ to temperature change (Figure 3 b, $P < 0.05$). In addition, Zou et al.
(2018) showed that the soil fluxes of $CO_2$ increased exponentially with increasing
temperature, but warming decreased the temperature sensitivity by 23% in the
grassland. Furthermore, Natali et al. (2011) also confirmed that, compared with the
$CO_2$ flux in the growing season, the $CO_2$ flux in the nongrowing season was more
sensitive to the temperature increase.
Ecosystem $CH_4$ flux is the net result of $CH_4$ production and consumption,
occurring simultaneously under the action of methanogenic archaea and
methane-oxidizing bacteria (e.g., Mer and Roger, 2001). In this study, warming
increased $CH_4$ uptake in the growing season, but decreased $CH_4$ uptake in the
non-growing season in the alpine grassland, findings similar to those from other
grassland ecosystems (Lin et al., 2015; Wu et al., 2020; Zhu et al., 2015). Our results
also demonstrated that seasonally asymmetric warming did not significantly affect the
response rate of $CH_4$ uptake (Figure 3 d-f, $P > 0.05$). $CH_4$ flux depended on
temperature, pH, and the availability of substrate (e.g., Treat et al., 2015). The $CH_4$
uptake observed during the three growing season and non-growing season implied
that the alpine grassland soil could act as an atmospheric $CH_4$ sink, a finding which
agrees with the results of many previous studies in similar regions (Wei et al., 2015;
Zhao et al., 2017). Hu et al. (2016) suggested that asymmetrical responses of $CH_4$
fluxes to warming and cooling should be taken into account when evaluating the
effects of climate change on $CH_4$ uptake in the alpine meadow on the Tibetan plateau.
Unlike $CH_4$ flux in alpine grasslands, Treat et al. (2018) confirmed that wetland was a
small $CH_4$ source in the non-growing season, whereas uplands varied from $CH_4$ sinks
to $CH_4$ sources. The latest research confirmed that warming in the Arctic had become
more apparent in the non-growing season than in the typical growing season (Bao et
al., 2020). Hereby, Bao et al. (2020) found that the $CH_4$ emissions during the spring
thaw and the autumn freeze contributed approximately one-quarter of the annual total
$CH_4$ emissions. That experimental warming is stimulating soil $CH_4$ uptake in the
growing season implies that the grasslands of the Bayinbuluk may have the potential
to remove more $CH_4$ from the atmosphere under future global warming conditions.
Furthermore, with the increased variation in soil temperature, the response rate of
$N_2O$ emission gradually decreased under AW treatment (Figure 3 g, $P < 0.05$). The
response of $N_2O$ emission to temperature increase was limited by the warming that
occurred throughout the year. However, $N_2O$ emission peaks were displayed during
the freeze–thaw periods (e.g., May 2017, June 2018 and April 2019). Warming
increased $N_2O$ emissions in the thawing period due to disruption of the gas diffusion
barrier and greater C and N availability for microbial activity (Nyborg et al., 1997).
Wagner-Riddle et al. (2017) also demonstrated that the magnitude of the
freeze/thaw-induced $N_2O$ emissions was associated with the number of days with soil
temperatures below $0^oC$. Pärn et al. (2018) found that $N_2O$ emission from organic
soils increases with rising soil $NO_3^-$, follows a bell-shaped distribution with soil
moisture. Another study has shown that a whole‑year warming treatment
significantly increased $N_2O$ emissions, but daytime, night-time or short‑season
warming did not have significant effects (Li et al., 2020). In addition, Cantarel et al.
(2010) suggested that the $N_2O$ flux from cool and upland grasslands may be driven
primarily by response to elevated temperature under projected future climate
conditions.
**5. Conclusions**
In summary, the effect of seasonally asymmetrical warming on $Re$ and $N_2O$
emission was obvious, unlike the situation with $CH_4$ uptake. The $Re$ and $N_2O$
emission were able to adapt to continuous warming, resulting in a reduced response
rates of the $Re$ and $N_2O$ emission to temperature increase. Warming in the
non-growing season increased the temperature dependence of the $Re$. Thus, we
believe that the study of climate change should pay greater attention to warming in the
non-growing season, to avoid underestimating the greenhouse effect on $Re$ in alpine
grasslands.
**Data availability**
The measured $CO_2$, $CH_4$ and $N_2O$ fluxes and soil temperature and soil water
content data are available in Zenodo (http://doi.org/10.5281/zenodo.4244207).
**Author contributions**
GYM, LYY and MA conceive the research question, designed the study approach,
led the field survey, ensured data curation and conducted formal analysis. YP and
LKH assisted with data collection and analysis. GYM wrote the first draft of the paper,
and all co-authors contributed to writing review and editing.
**Competing interests**
The authors declare that they have no conflicts of interest.
**Acknowledgments**
This work was supported by the NSFC Program (41603084, 41703131,

296 41673079).

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

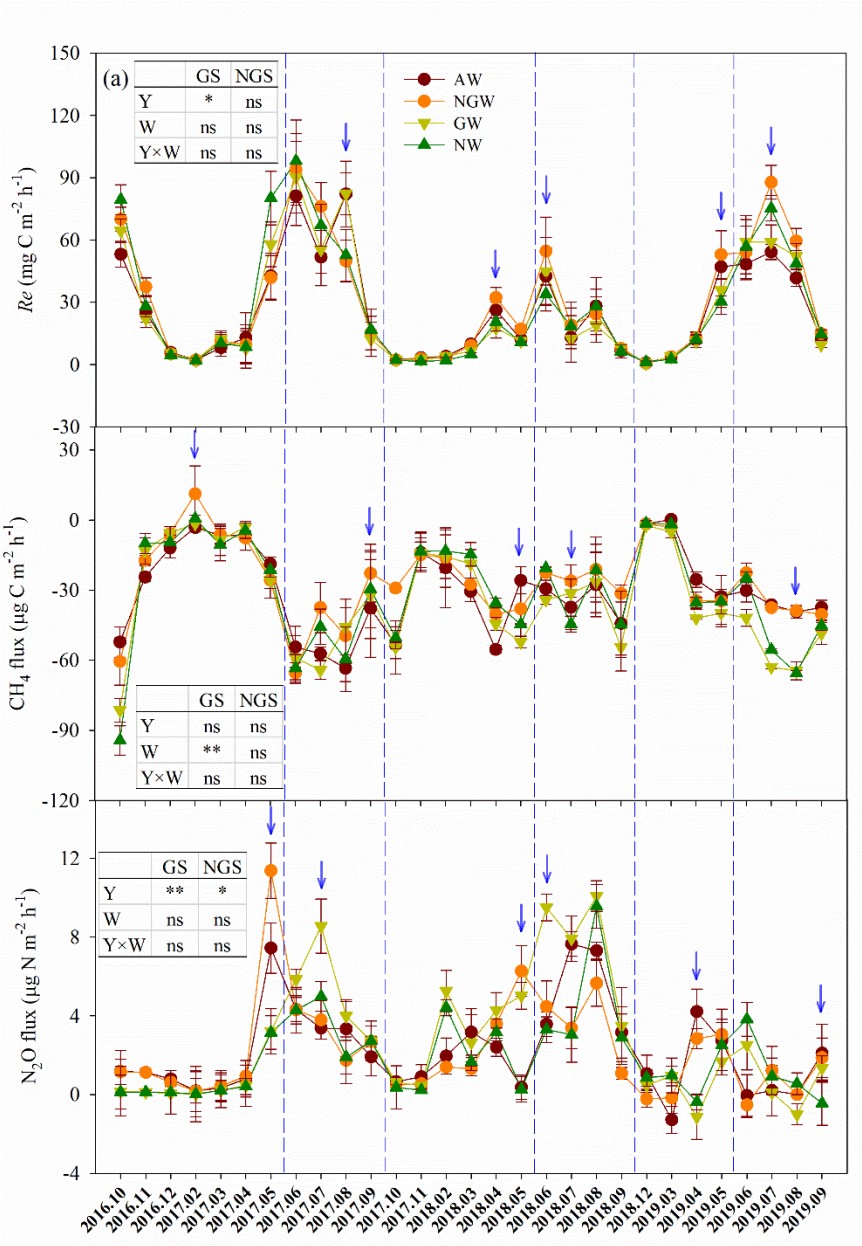


Figure 1 Monthly variation of a). ecosystem respiration (*Re*), b). CH₄ uptake and c).

N₂O emissions under the four treatments from October 2016 to September 2019. AW,

warming throughout the year; NGW, warming in the non-growing season only; GW,

warming in the growing season only; NW, non-warming. The blue arrows indicate

warming effects. The data points represent mean ± standard error, SE. The tables

illustrate the tests of significance for year (Y) and warming (W) on *Re*, CH₄ uptake

and N₂O emission by two-way repeated-measures analysis of variance (ANOVA) in

the growing season (GS) and the non-growing season (NGS), respectively; $^{*}P < 0.05$;

$^{**}P < 0.01$; ns, non-significant.

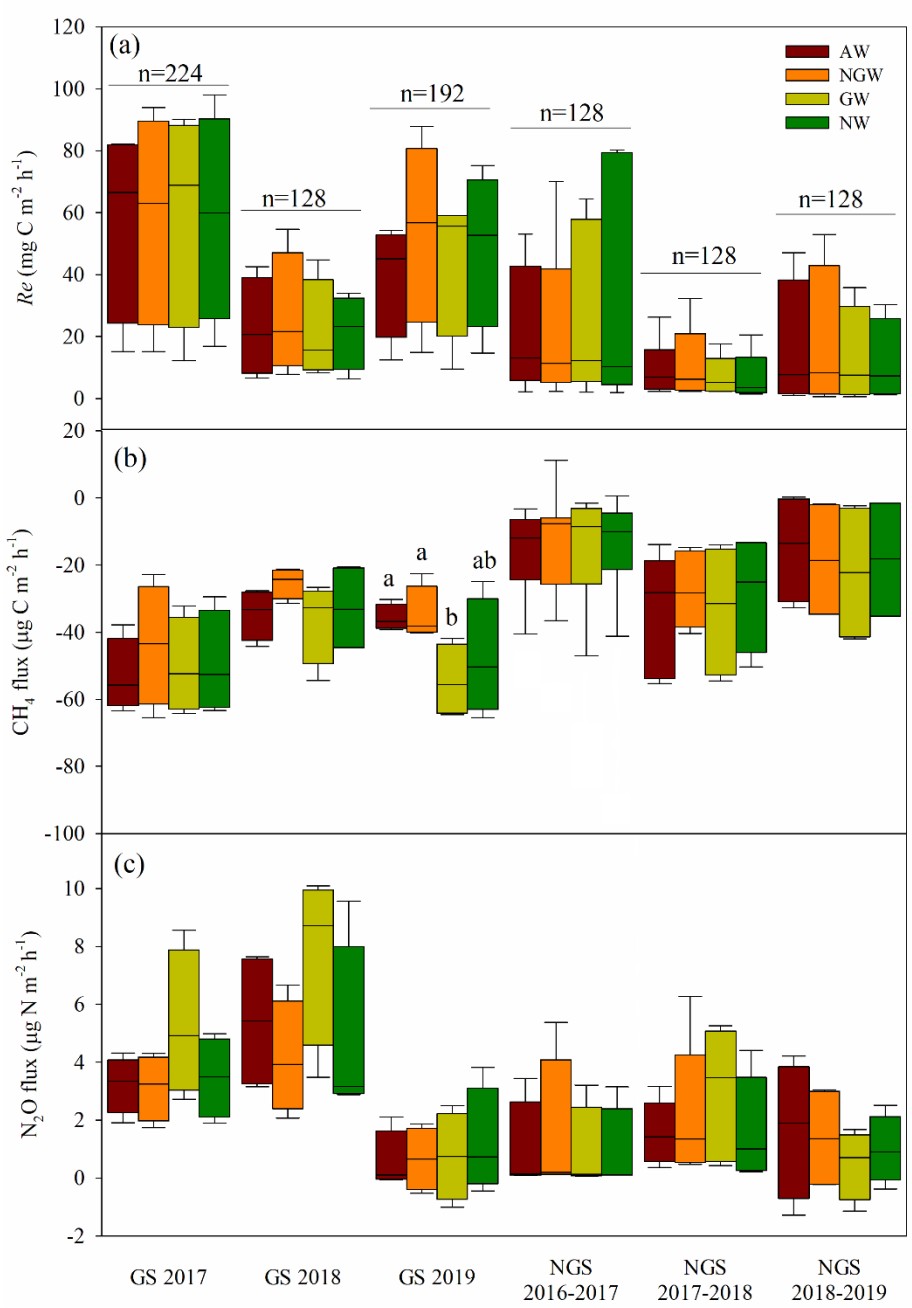

Figure 2 Boxplot presentation of variations in ecosystem respiration ($Re$), $CH_4$ uptake, and $N_2O$ emission under four treatments in the growing season and non-growing season from October 2016 to September 2019. The median is represented by the black line in the box. The box (the interquartile range) represents

the middle 50% of the data, whereas the whiskers represent the ranges for the bottom
25% and the top 25% of the data values, excluding outliers. GS, growing season; NGS,
non-growing season; AW, warming throughout the year; NGW, warming in the
non-growing season only; GW, warming in the growing season only; NW,
non-warming. No significant differences among AW, NGW, GW, and NW were
reported from ANOVA; data points are the mean ± standard error. One-way ANOVA
results of $Re$, $CH_4$ uptake and $N_2O$ emissions among the four warming treatments
were not significant, except that the $CH_4$ uptake in the GS 2019 under the GW
treatment was significantly higher than that of AW and NGW treatment ($P < 0.05$).

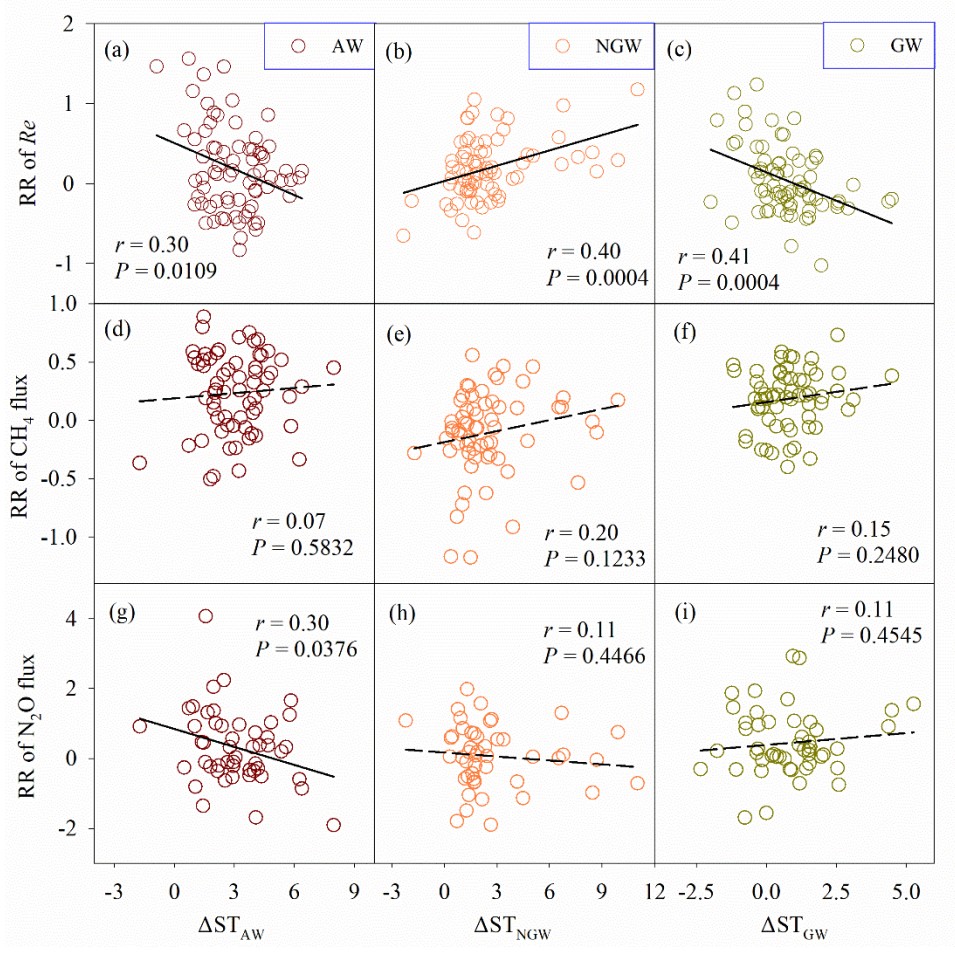


Figure 3 Response (presented by linear correlation) of variation in ecosystem
respiration ($Re$), $CH_4$ uptake, and $N_2O$ emission to changes in soil temperature under
AW, NGW and GW conditions in the alpine grassland, from 2016 to 2019. RR, the
natural logarithm of the response ratio of the mean value of the chosen variable in the
warming group to that in the control (NW) group. $\Delta ST_{AW}$, soil temperature of AW
minus that of NW; $\Delta ST_{CW}$, soil temperature of NGW minus that of NW; $\Delta ST_{WW}$, soil
temperature of GW minus that of NW; AW, warming throughout the year; NGW,
warming in the non-growing season only; GW, warming in the growing season only;
NW, non-warming.

| | $Re$ | | | $CH_4$ flux | | | $N_2O$ flux | | |
|---|---|---|---|---|---|---|---|---|---|
| | a | c | b | | | | | | |
| NGW-NGS % | **41.6** | 0.8 | -1.6 | **75.0** | -4.1 | 0.8 | **43.8** | -1.4 | -1.9 |
| NGW-GS % | 6.4 | 6.3 | 9.0 | -2.9 | 0.2 | -2.7 | 1.3 | 4.0 | -0.3 |
| GW-NGS % | 0.7 | **36.5** | **22.2** | **51.3** | 7.4 | 0.9 | **29.6** | 10.2 | -2.0 |
| GW-GS % | **22.6** | -12.4 | **23.4** | -2.6 | 0.4 | -2.4 | 3.8 | 0.9 | <0.1 |
| AW-AY % | 9.5 | **22.3** | 10.1 | 15.3 | 6.2 | -0.9 | 7.7 | 4.5 | -1.9 |
| NW-AY % | 7.6 | **26.7** | 5.0 | 18.5 | 4.7 | -0.9 | **21.5** | -3.7 | 3.5 |


Figure 4 Influence of soil temperature and soil moisture on ecosystem respiration
(*Re*), CH₄ uptake, and N₂O emission by variation-partitioning analysis under four
treatments in the growing season and non-growing season. a, Single effect of soil
temperature (%); b, single effect of soil moisture (%); c, joint effects of soil
temperature and moisture (%); NGW-NGS, greenhouse gas fluxes in non-growing
season under non-growing season warming treatment; NGW-GS, greenhouse gas
fluxes in growing season under non-growing season warming treatment; GW-NGS,
greenhouse gas fluxes in non-growing season under growing season warming
treatment; GW-GS, greenhouse gas fluxes in growing season under growing season
warming treatment; AW-AY, annual greenhouse gas fluxes under annual warming
treatment; NW-AY, annual greenhouse gas fluxes without warming.