# Peer review of "Different responses of ecosystem CO2 and N2O emissions and CH4 uptake to"

_Biogeosciences, 2020_

## Short Comment (SC1) · 1 Dec 2020

This manuscript describes the response of GHGs emissions to seasonally asymmetric warming in an alpine grassland of Tianshan Mountains. It is an interesting topic to understand carbon and nitrogen cycles with increasing temperature. The manuscript is well written and concise. The experiment is well designed and conducted. I suggest this manuscript could be accepted after some minor revisions. Introduction: the research advances of responses of CO2, CH4 and N2O fluxes to seasonally asymmetric warming is very limited, more contents could be added especially in grassland ecosystem. Authors quoted many IPCC results about warming and its effect on GHGs

fluxes, which need to be summarized. Materials and methods: air temperature and precipitation data of growing season and non-growing season could not be found, which are important to explain the effect of seasonally asymmetric warming on GHGs flux. Discussion: please delete figure 2 and P < 0.05 or P > 0.05. The manuscript do not research the response of GHGs to daytime, nighttime or short-season warming, please delete it. Conclusions: please add the responses of CH4 and N2O fluxes to warming in the study.

---

## Referee Comment (RC1) · Anonymous Referee #1 · 15 Dec 2020

I have reviewed this paper by Gong et al. This study revealed the effect of seasonally asymmetric warming on greenhouse gas fluxes in alpine grassland, and it advances our understanding of warming effects on greenhouse gas fluxes. I think there are a few minor issues that could be improved Before it could be accepted for publication. 1) The authors should focus more on the mechanisms behind the different responses of greenhouse gas fluxes to seasonally asymmetric warming. 2) There are a few minor things needed to be revised throughout the manuscript. For example, Line 16: "greenhouse gas flux" should be changed to "greenhouse gas fluxes"; Line 154: "Figure 2" should be "Fig. 2" to be consistent with other places.

---

## Referee Comment (RC2) · Anonymous Referee #2 · 18 Dec 2020

General comments

The manuscript "Different responses of CO2, CH4, and N2O fluxes to seasonally asymmetric warming in an alpine grassland of Tianshan Mountains" by Gong et al. describes the effects of seasonal warming (growing season warming, non-growing season warming, annual warming) on CO2, CH4, and N2O fluxes during 3 years at an alpine grassland site in the southern Tianshan mountains, China.

I value the authors efforts to collect multi-year, and year-round data with manual chamber measurements. While I find evaluating the response of GHG fluxes to changes in temperature during different seasons highly important (especially the non-growing

season), the manuscript in its current state has several shortcomings, which I have outlined in my specific comments and line edits below.

Specific comments

1) The authors discuss $CO_2$ fluxes, however, it is unclear which component of the $CO_2$ flux has been measured. It would seem that ecosystem respiration was measured, which should be stated clearly throughout the manuscript. Temperature is a well known control on respiration, but based on this study the authors can not draw conclusions on the effect of temperature on net $CO_2$ exchange without accounting for photosynthesis. I suggest revising the manuscript text accordingly. 2) To understand the observed interannual variations in GHG exchange in response to temperature it would be important to include information on other climate parameters as well, especially interannual variations in precipitation, soil moisture and or water table. If this data is available, I highly recommend including it and to add discussion on this topic. 3) Adding discussion on whether the warming treatment with OTCs impacted other environmental variables (such as soil moisture, snow depth) would be needed in order to assess the effectiveness of the warming treatment and validity of results. Shortly reporting results on the achieved temperature increase in the warmed plots compared to control treatments in the results section would be helpful as well (this is only shown in the supplementary material Fig S1). 4) Introduction as well as discussion and conclusion remain rather superficial. This manuscript would greatly benefit from some streamlining, clearly stating the objectives and relevance of this study, and a more thorough literature review and comparison to other studies. For example, this study reports rather large $CH_4$ uptake rates. How do these rates compare to what is observed in other studies from similar ecosystems? Based on the findings of this study, can larger-scale conclusions be drawn on what impact warming will have on $CH_4$ uptake in these ecosystems? The study also reports all three GHGs, which is a strength of this study, as measurements of $N_2O$ fluxes in particular are rare in colder climates. The authors could highlight this in their study, and provide some comparison to other studies. And while investigating the effect of temperature on fluxes was clearly the aim of this study, some acknowledgement of drivers of GHG fluxes other than temperature would be useful to include. 5) The discussion is rather short and exclusively focuses on the reponse rate of the three gases to temperature, while some of the rather interesting key findings of this study are not addressed (such as for example increasing annual CH4 uptake with warming, or increasing N2O emissions with warming during the non-growing season). It also looks like the site displayed emission peaks of N2O during the shoulder periods, especially spring, which might be an important point to mention in the discussion (see for example: Wagner-Riddle et al. (2017). Globally important nitrous oxide emissions from croplands induced by freeze–thaw cycles. Nature Geoscience 10(4):279-83).

Line edits

Abstract: L13-14: specify whether CO2 fluxes are ecosystem respiration or net ecosystem exchange, not clear if the reported numbers are net CO2 losses to the atmosphere. Also, do these numbers represent the total range of fluxes between 2016 and 2019? It might be more meaningful to present growing season as well as annual mean or median fluxes in the abstract, as fluxes, especially for CH4 and N2O, are highly variable. L14-15: this is counter intuitive and does not match with what is shown in supplementary figure S3 (where CO2 and N2O fluxes show a clear positive correlation with soil temperature, and CH4 uptake increases with increases temperature). Please rephrase. L16-18: "the variation in GHG flux under seasonally asymmetric warming was different between the growing season and the non-growing season": this statement is vague, please be more specific and state clearly what was observed. L18-24: a short explanation of the term "response rate" might be needed in this context. L24-27: A clear summary statement with the specific implications of this study would be needed here.

Introduction: L38: daytime/nighttime differences are not addressed in this study. Consider removing from the introduction or add results and discussion to address this issue. L34-36: Yes, but I would advise caution with this statement (considering the larger than average warming in higher latitudes and Arctic amplification). L39: 3rd

assessment report. L48: delete "in the atmosphere" L47-49: some rephrasing might be needed (considering that water vapour is a major GHG present in Earth's atmopshere as well). L47-50: a general statement with some background information on the influence of temperature on GHG production and emissions would be useful for the reader in this context. L50: Please check those % numbers (radiative forcing of $CO_2$ is larger than that of $CH_4$). L50-52: Not sure whether this statement is correct. At least in northern or high-elevation regions with less accessible sites, warming treatments are often conducted during the summer months. Some rephrasing might be needed. L55: please specify whether "$CO_2$ flux" in this context refers to increased $CO_2$ emissions or increased net uptake, and under what conditions this increase occurred (warming treatment or naturally warmer summer?). L56-57: for simplicity, replace "the effect of increased temperature in winter" with "the effect of winter warming". L56-58: This sentence is not quite clear. Does this mean winter warming did not affect growing season $CO_2$ fluxes, but winter warming did increase $CO_2$ fluxes during the nongrowing season? Consider rephrasing. L58-60: replace "absorption" with "uptake. L62: start sentence with "A study by xx (2012) in an alpine grassland ecosystem showed. . ." L69: consider added examples for biotic and abiotic factors here. L75-81: This is rather vague and I suggest to be more specific and clearly state the overall aim of this study.

Methods: L87: Is permafrost present at the site? If yes, it would be useful to add this information. L93: in addition to soil, vegetation and temperature it would be useful to add some information related to typical soil moisture/water table levels at the site, since that is important for discussing GHG exchange and observed interannual differences. L95: please provide description and dimensions of open-top chambers. L101-108: please provide some more details regarding flux measurement and analysis. E.g., were pre-installed collars used for the flux measurement? Were the chambers equipped with a fan and pressure equilibration tube? Were chambers transparent or opaque? Please also mention the sign convention in this context (i.e. positive fluxes = emissions?). L105: please add the total number of sampling times. L105-108: What quality criteria were used to accept or reject fluxes? E.g. r2, RMSE, minimum number of sampling

points during one flux measurements? This information is not provided in the cited reference (Chen et al 2013).

Results: L124-125: In addition to the range, it would be useful to provide mean/median values here. I also suggest to add a few sentences describing the general pattern of fluxes as this site, before presenting the treatment results, to provide the reader with a general overview (for example stating that the site acted as a net $CH_4$ sink, with negligible $CH_4$ emissions, and small $N_2O$ source). L131-136: talking about an increase and decrease in $CH_4$ flux is slightly confusing in this context, as the authors mainly observed $CH_4$ uptake. The 6.4% increase in $CH_4$ flux in the AW treatment that the authors report here is in fact an increase in $CH_4$ uptake, according to Fig. S2. This would mean a decrease in $CH_4$ from an atmospheric point of view, and I suggest to rephrase this whole section accordingly. To avoid confusion, it might be useful to refer to uptake and emissions, rather than fluxes, throghout the manuscript. L126-140: are all these reported %changes significant (standard errors seem rather large)? It would be useful to add information regarding statistical significance in Fig S2 and manuscript text. L126 and throughout: please specify which $CO_2$ flux component is discussed (see specific comments above). L143: delete "extremely". L141-144: Do the authors mean interannual differences between growing seasons? L141-145: Generally, this section is not very clear and would benefit from some rephrasing. It would be important to state that interannual differences were larger than impact of warming treatment (for $CO_2$ and $N_2O$) according to Fig. 1, whereas warming treatment had a significant impact on $CH_4$ fluxes. L147-148: I suggest simplifying "$CH_4$ flux showed significantly decreasing trends with increasing soil temperature" to something like "we observed increasing $CH_4$ uptake with increasing soil temperature".

Discussion and conclusion: Please see my specific comments above regarding the discussion section, as well as: L166-171: It would be useful to include some background information on mechanisms behind $CH_4$ fluxes for the reader, i.e. when do emissions occur, what conditions promote $CH_4$ uptake, why would temperature increase $CH_4$

uptake, etc. L176-177: for the nongrowing season contribution of CH4 fluxes a comparison to other ecosystems would be useful; see for example Treat, C.C., Bloom, A.A. and Marushchak, M.E., 2018. Nongrowing season methane emissions–a significant component of annual emissions across northern ecosystems. Global change biology, 24(8), pp.3331-3343. L178-186: please see my specific comment regarding discussion of other environmental variables besides temperature. N2O in particular is rarely depend on just on variable, and the effect of temperature may often be masked by other variables such as water table, and mineral nitrogen availability. This may require at least short mention in the discussion section. See for example Pärn, J. et al. 2018. Nitrogen-rich organic soils under warm well-drained conditions are global nitrous oxide emission hotspots. Nature communications, 9(1), pp.1-8.

Figures: Fig. 2: please add r2 and P-values for all figure panels (even for non-significant relationships). Fig S2: please add number of measurement times for growing season / non-growing season mean. Also, please specify in y-axis or figure caption which component of the CO2 flux is shown (ER?). As panel b shows CH4 uptake, I suggest to flip the y-axis, showing zero on top and negative values at the bottom. Overall, I would suggest to use boxplots (including quartile ranges and outliers) rather than barplots in this figure, to capture the full range of fluxes, as it would be important to show e.g. also the occurrence of emissions (for CH4) or uptake (for N2O). The authors may also consider moving this figure into the main text.

---

## Author Comment (AC1) · 11 Jan 2021

Thank you for your comments and the reviewers' comments concerning our manuscript entitled " Different responses of CO2, CH4, and N2O fluxes to seasonally asymmetric warming in an alpine grassland of Tianshan Mountains" (MS No.: bg-2020-396). Those comments are all valuable and very helpful for revising and improving our paper, as well as the important guiding significance to our researches. We have studied those comments carefully and have made a point to point reply and correction. Revised portion are marked in red color in this manuscript. Specific corrections and responds to the reviewer's comments are listed as follows:

[Figure]

Comments to the Author: # the comment by Y.G. Du This manuscript describes the response of GHGs emissions to seasonally asymmetric warming in an alpine grassland of Tianshan Mountains. It is an interesting topic to understand carbon and nitrogen cycles with increasing temperature. The manuscript is well written and concise. The experiment is well designed and conducted. I suggest this manuscript could be accepted after some minor revisions. Introduction: the research advances of responses of CO2, CH4 and N2O fluxes to seasonally asymmetric warming is very limited, more contents could be added especially in grassland ecosystem. Authors quoted many IPCC results about warming and its effect on GHGs fluxes, which need to be summarized. Response: Thank you for your precise comment for the Introduction. We have added to the latest research on the effect of warming on greenhouse gas flux in grassland ecosystems. Add the following: "A recent study showed that seasonal variations in carbon flux were more related to air temperature in the meadow steppe (Zhao et al., 2019). Another study found that experimental warming enhanced CH4 uptake in the relatively arid alpine steppe, but had no significant effects on CH4 emission in the moist swamp meadow (Li et al., 2020). Wu et al. (2020) also showed that the warming did not significantly affect soil CH4 uptake fluxes in the alpine meadow of the Tibetan Plateau. Furthermore, a global meta-analysis (Wang et al., 2019) showed that experimental warming stimulates C fluxes in grassland ecosystems, and the response of C fluxes to warming strongly varies across the different grassland types, with higher warming responses in cold than in temperate and semi‐arid grasslands. Across the data set, Li et al. (2019) demonstrated that whole day or whole year warming treatment significantly enhanced N2O emissions, but daytime, nighttime or short season warming did not have significant effects." See L175-187 of the revised manuscript. We also summarized the IPCC results about warming and its effect on GHGs fluxes. The revised content is as follows: "The 3th and 4th assessment report of the Inter-Governmental Panel on Climate Change (IPCC) proposed that, against the backdrop of global warming, the temperature change will show that the warming amplitude in winter is greater than that in summer, and the warming amplitude at high latitude is greater than that at

low latitude, and confirmed that the warming shows asymmetric trends on a seasonal scale (Easterling et al., 1997; IPCC, 2001; IPCC, 2007)." See L86-91 of the revised manuscript.

Materials and methods: air temperature and precipitation data of growing season and non-growing season could not be found, which are important to explain the effect of seasonally asymmetric warming on GHGs flux. Response: Thank you for your comment for the Materials and methods. We have presented air temperature and precipitation data through tow figures. And described the Figure S3 and S4 in the revised manuscript.

Figure S3 Variation in air temperature (inside the OTC, 50cm above the ground) under four treatments in alpine grassland from October 2016 to September 2019. GS, growing season; NGS, non-growing season; AW, warming throughout the year; NGW, warming in nongrowing season only; GW, warming in growing season only; NW, non-warming. Significant differences among AW, NGW, GW and NW from analysis of variance (ANOVA) are denoted as bars within the same season with different lowercase letters, P < 0.05; data points are the mean ± standard error.

Figure S4 Variation in precipitation in the alpine grassland from October 2016 to September 2019. GS, growing season; NGS, non-growing season.

Discussion: please delete figure 2 and P < 0.05 or P > 0.05. The manuscript do not research the response of GHGs to daytime, nighttime or short-season warming, please delete it. Response: Thank you for precise comment for the Discussion. However, we disagree with this comment. Figure 2 shows the highlights of this manuscript: "Response of variations in CO2, CH4, and N2O fluxes to changes in soil temperature under AW, NGW and GW conditions in the alpine grassland, from 2016 to 2019." Figure 2 does not mention what the comments suggest: "the response of GHGs to daytime, nighttime or short-season warming"

Conclusions: please add the responses of CH4 and N2O fluxes to warming in the

study. Response: Thank you for your precise comment for the Conclusions. We have revised the conclusion as "In summary, the effect of seasonally asymmetrical warming on $CO_2$ and $N_2O$ fluxes were obvious, but not $CH_4$ flux, with the $CO_2$ and $N_2O$ fluxes being able to adapt to continuous warming, resulting in a reduced response rate of the $CO_2$ and $N_2O$ fluxes to temperature increase. Warming in the nongrowing season increased the temperature dependence of the $CO_2$ flux. Thus, we believe that the study of climate change should pay greater attention to warming in the nongrowing season, so as not to underestimate the greenhouse effect of the $CO_2$ flux in alpine grasslands." See L535-541 of the revised manuscript.
* * *
**Different responses of ecosystem respiration, CH₄ uptake, and N₂O emissions to
seasonally asymmetric warming in an alpine grassland of Tianshan Mountains**

[Figure]

Figure S3 Variation in air temperature (inside the OTC, 50cm above the ground) under four treatments in alpine grassland from October 2016 to September 2019. GS, growing season; NGS, non-growing season; AW, warming throughout the year; NGW, warming in nongrowing season only; GW, warming in growing season only; NW, non-warming. Significant differences among AW, NGW, GW and NW from analysis of variance (ANOVA) are denoted as bars within the same season with different lowercase letters, $P < 0.05$; data points are the mean ± standard error.

**Fig. 1.**

---

## Author Comment (AC2) · 11 Jan 2021

Anonymous Referee #1 I have reviewed this paper by Gong et al. This study revealed the effect of seasonally asymmetric warming on greenhouse gas fluxes in alpine grassland, and it advances our understanding of warming effects on greenhouse gas fluxes. I think there are a few minor issues that could be improved Before it could be accepted for publication. 1) The authors should focus more on the mechanisms behind the different responses of greenhouse gas fluxes to seasonally asymmetric warming. Response: Thank you for your precise comment. We revised the manuscript. L421-438, "Ecosystem CH4 flux is the net result of CH4 production and consumption occur

simultaneously under the action of methanogenic archaea and methane-oxidizing bacteria (e.g., Mer and Roger, 2001; Blodau, 2002; Galbally et al., 2008). CH4 fluxes are dependent on temperature, pH, and the availability of substrate (e.g., Moore & Dalva, 1997; Treat et al., 2015). For example, lower levels of soil moisture would decrease C release, indicating that drier soils are the major CH4 sink (Denman et al., 2007). The CH4 uptake observed during the three growing and non-growing seasons implied that the alpine grassland soil could act as an atmospheric CH4 sink, which agrees with the findings of many previous studies in similar region (Li et al., 2015; Wei et al., 2015; Zhao et al., 2017; Wu et al., 2020). As well as, our results demonstrated that warming increased CH4 uptake in the growing season, but decreased CH4 uptake in the non-growing season in the alpine grassland, similar to the results from other grassland ecosystems (Lin et al., 2015; Zhu et al., 2015; Wu et al., 2020). Moreover, a warming-induced soil moisture decrease may be a potential mechanism for the positive influence of warming on soil CH4 uptake in the alpine grassland (Lin et al., 2015). Our results also demonstrated that seasonally asymmetric warming did not significantly affect the response rate of CH4 uptake to temperature increase (Figure 2 d-f, P > 0.05)." L441-502, "Unlike CH4 fluxes in alpine grasslands, Treat et al. (2018) confirmed that nongrowing season wetland were small CH4 sources, and uplands varied from CH4 sinks to CH4 sources." L515-518, "As well as N2O emissions was positively related to soil temperature, Pärn et al. (2018) found that N2O emission from organic soils increases with rising soil NO3-, follows a bell-shaped distribution with soil moisture." L512-518, "However, our results displayed emissions peaks of N2O during the freeze–thaw periods (e.g., May 2017, June 2018 and April 2019). Warming increased N2O emissions in the thawing period owning to disrupt the gas diffusion barrier and greater C and N availability for microbial activity (Nyborg et al., 1997). Wagner-Riddle et al. (2017) also confirmed that the magnitude of freeze–thaw-induced N2O emissions was related to the number of days with soil temperatures below 0 oC"

2) There are a few minor things needed to be revised throughout the manuscript. For example, Line 16: "greenhouse gas flux" should be changed to "greenhouse gas

fluxes"; Line 154: "Figure 2" should be "Fig. 2" to be consistent with other places. Response: Thank you for your precise comment. Line 16: we revised as "greenhouse gas fluxes". See L20 of the revised manuscript.
* * *

---

## Author Comment (AC3) · 11 Jan 2021

Anonymous Referee #2 General comments The manuscript "Different responses of CO2, CH4, and N2O fluxes to seasonally asymmetric warming in an alpine grassland of Tianshan Mountains" by Gong et al. describes the effects of seasonal warming (growing season warming, non-growing season warming, annual warming) on CO2, CH4, and N2O fluxes during 3 years at an alpine grassland site in the southern Tianshan mountains, China. I value the authors efforts to collect multi-year, and year-round data with manual chamber measurements. While I find evaluating the response of GHG fluxes to changes in temperature during different seasons highly important (es-

pecially the non-growing season), the manuscript in its current state has several short-comings, which I have outlined in my specific comments and line edits below. Specific comments 1) The authors discuss CO2 fluxes, however, it is unclear which component of the CO2 flux has been measured. It would seem that ecosystem respiration was measured, which should be stated clearly throughout the manuscript. Temperature is a well known control on respiration, but based on this study the authors can not draw conclusions on the effect of temperature on net CO2 exchange without accounting for photosynthesis. I suggest revising the manuscript text accordingly. Response: As the anonymous referee' comment, we're measuring ecosystem respiration. We revised the manuscript text accordingly. See the revised manuscript.

2) To understand the observed interannual variations in GHG exchange in response to temperature it would be important to include information on other climate parameters as well, especially interannual variations in precipitation, soil moisture and or water table. If this data is available, I highly recommend including it and to add discussion on this topic. Response: Thank you for your precise comment. We plot the changes of precipitation, air temperature and soil moisture, and analyze the influence of soil temperature and moisture on greenhouse gas fluxes and their association effect through variance decomposition. we do not have the monitoring data of the groundwater level in Bayanbulak Grassland. We will carry out the work in the future. It was discussed and analyzed in the revised manuscript. As shown in the figure below:

Figure S2 Variation in soil water content (at 10-cm depth) under four treatments in alpine grassland from June 2017 to September 2019. GS, growing season; NGS, non-growing season; AW, warming throughout the year; NGW, warming in nongrowing season only; GW, warming in growing season only; NW, non-warming. Significant differences among AW, NGW, GW and NW from analysis of variance (ANOVA) are denoted as bars within the same season with different lowercase letters, P < 0.05; data points are the mean $\pm$ standard error.

Figure S3 Variation in air temperature (inside the OTC, 50cm above the ground) under four treatments in alpine grassland from October 2016 to September 2019. GS, growing season; NGS, non-growing season; AW, warming throughout the year; NGW, warming in nongrowing season only; GW, warming in growing season only; NW, non-warming. Significant differences among AW, NGW, GW and NW from analysis of variance (ANOVA) are denoted as bars within the same season with different lowercase letters, $P < 0.05$; data points are the mean $\pm$ standard error.

Figure S4 Variation in precipitation in the alpine grassland from October 2016 to September 2019. GS, growing season; NGS, non-growing season.

Figure S6 Influence of soil temperature and soil moisture on ecosystem respiration (Re), CH4 uptake, and N2O emission by variation-partitioning analysis under four treatments in the growing season and nongrowing season. a, single effect of soil temperature (%); b, single effect of soil moisture (%); c, joint effects of soil temperature and moisture (%); NGW-NGS, greenhouse gas fluxes in non-growing season under non-growing season warming treatment; NGW-GS, greenhouse gas fluxes in growing season under non-growing season warming treatment; GW-NGS, greenhouse gas fluxes in non-growing season under growing season warming treatment; GW-GS, greenhouse gas fluxes in growing season under growing season warming treatment; AW-AY, annual greenhouse gas fluxes under annual warming treatment; NW-AY, annual greenhouse gas fluxes without warming.

3) Adding discussion on whether the warming treatment with OTCs impacted other environmental variables (such as soil moisture, snow depth) would be needed in order to assess the effectiveness of the warming treatment and validity of results. Shortly reporting results on the achieved temperature increase in the warmed plots compared to control treatments in the results section would be helpful as well (this is only shown in the supplementary material Fig S1). Response: Thank you for your precise comment. Yes, warming treatment with OTCs impacted soil moisture and snow depth. The change of snow depth mainly affects the soil moisture, therefore, we focused on the analysis of how warming affects the greenhouse gas flux by affecting soil moisture

(Figure S6). We added the information in the revised manuscript.

4) Introduction as well as discussion and conclusion remain rather superficial. This manuscript would greatly benefit from some streamlining, clearly stating the objectives and relevance of this study, and a more thorough literature review and comparison to other studies. For example, this study reports rather large CH4 uptake rates. How do these rates compare to what is observed in other studies from similar ecosystems? Based on the findings of this study, can larger-scale conclusions be drawn on what impact warming will have on CH4 uptake in these ecosystems? The study also reports all three GHGs, which is a strength of this study, as measurements of N2O fluxes in particular are rare in colder climates. The authors could highlight this in their study, and provide some comparison to other studies. And while investigating the effect of temperature on fluxes was clearly the aim of this study, some acknowledgement of drivers of GHG fluxes other than temperature would be useful to include. Response: Thank you for your precise comment. We revised the manuscript and added the information of your comments. For example, L86-112, "The 3rd and 4rd assessment report of the Inter-Governmental Panel on Climate Change (IPCC) proposed that, against the backdrop of global warming, the temperature change will show that the warming amplitude in winter is greater than that in summer, and the warming amplitude at high latitude is greater than that at low latitude, and confirmed that the warming shows asymmetric trends on a seasonal scale (Easterling et al., 1997; IPCC, 2001; IPCC, 2007)." L116-121, "Experimental warming is known to influence the Re, CH4 uptake and N2O emission (Parn et al., 2018; Treat et al., 2018; Wang et al., 2019). Information on Re, CH4 uptake and N2O emission and their sensitivity to warming, will enhance our understanding of ecosystem C and N cycling processes and improve our predictions of the response of ecosystems to global climate change (Li et al., 2020; Wang et al., 2019)." L124-129, "A study of the Alaskan tundra found that summer warming (open-top chambers to increase growing season air temperatures) significantly increased the growing season ecosystem respiration by about 20 % (Natali et al., 2011). Compared with the slight effect of winter warming on the CO2 fluxes in the growing season, warming increased CO2 fluxes during the snow-covered period by more than 50% (Natali et al., 2011)." L175-187, "A recent study showed that seasonal variations in carbon flux were more related to air temperature in the meadow steppe (Zhao et al., 2019). Another study found that experimental warming enhanced CH4 uptake in the relatively arid alpine steppe, but had no significant effects on CH4 emission in the moist swamp meadow (Li et al., 2020). Wu et al. (2020) also showed that the warming did not significantly affect soil CH4 uptake fluxes in the alpine meadow of the Tibetan Plateau. Furthermore, a global meta-analysis (Wang et al., 2019) showed that experimental warming stimulates C fluxes in grassland ecosystems, and the response of C fluxes to warming strongly varies across the different grassland types, with higher warming responses in cold than in temperate and semi‐arid grasslands. Across the data set, Li et al. (2019) demonstrated that whole day or whole year warming treatment significantly enhanced N2O emissions, but daytime, nighttime or short season warming did not have significant effects." L192-203, "For example, over longer time periods of warming, accelerated decomposition and increased plant N uptake may decrease soil organic C and N pools (Wu et al., 2012), and the microbial community with variable C use efficiency may reduce the temperature sensitivity of heterotrophic respiration (Zhou et al., 2011)." L207-210, "Therefore, we hypothesize that warming in non-growing season will stimulate greenhouse, e gas flux (especially during non-growing season) in alpine steppe. However, continuous warming throughout the year and during the growing season will weaken the sensitivity of greenhouse gas flux to warming." L421-438, "Ecosystem CH4 flux is the net result of CH4 production and consumption occur simultaneously under the action of methanogenic archaea and methane-oxidizing bacteria (e.g., Mer and Roger, 2001; Blodau, 2002; Galbally et al., 2008). CH4 fluxes are dependent on temperature, pH, and the availability of substrate (e.g., Moore & Dalva, 1997; Treat et al., 2015). For example, lower levels of soil moisture would decrease C release, indicating that drier soils are the major CH4 sink (Denman et al., 2007). The CH4 uptake observed during the three growing and non-growing seasons implied that the alpine grassland soil could act as an atmospheric CH4 sink, which agrees

with the findings of many previous studies in similar region (Li et al., 2015; Wei et al., 2015; Zhao et al., 2017; Wu et al., 2020). As well as, our results demonstrated that warming increased CH4 uptake in the growing season, but decreased CH4 uptake in the non-growing season in the alpine grassland, similar to the results from other grassland ecosystems (Lin et al., 2015; Zhu et al., 2015; Wu et al., 2020). Moreover, a warming-induced soil moisture decrease may be a potential mechanism for the positive influence of warming on soil CH4 uptake in the alpine grassland (Lin et al., 2015). Our results also demonstrated that seasonally asymmetric warming did not significantly affect the response rate of CH4 uptake to temperature increase (Figure 2 d-f, P > 0.05)." L441-502, "Unlike CH4 fluxes in alpine grasslands, Treat et al. (2018) confirmed that nongrowing season wetland were small CH4 sources, and uplands varied from CH4 sinks to CH4 sources." L515-518, "As well as N2O emissions was positively related to soil temperature, Pärn et al. (2018) found that N2O emission from organic soils increases with rising soil NO3-, follows a bell-shaped distribution with soil moisture."

5) The discussion is rather short and exclusively focuses on the reponse rate of the three gases to temperature, while some of the rather interesting key findings of this study are not addressed (such as for example increasing annual CH4 uptake with warming, or increasing N2O emissions with warming during the non-growing season). It also looks like the site displayed emission peaks of N2O during the shoulder periods, especially spring, which might be an important point to mention in the discussion (see for example: Wagner-Riddle et al. (2017). Globally important nitrous oxide emissions from croplands induced by freeze–thaw cycles. Nature Geoscience 10(4):279-83). Response: Thank you for your precise comment. We revise the sections according to your comments. See the revised manuscript, L421-438, "Ecosystem CH4 flux is the net result of CH4 production and consumption occur simultaneously under the action of methanogenic archaea and methane-oxidizing bacteria (e.g., Mer and Roger, 2001; Blodau, 2002; Galbally et al., 2008). CH4 fluxes are dependent on temperature, pH, and the availability of substrate (e.g., Moore & Dalva, 1997; Treat et al., 2015). For example, lower levels of soil moisture would decrease C release, indicating that drier soils

are the major CH4 sink (Denman et al., 2007). The CH4 uptake observed during the three growing and non-growing seasons implied that the alpine grassland soil could act as an atmospheric CH4 sink, which agrees with the findings of many previous studies in similar region (Li et al., 2015; Wei et al., 2015; Zhao et al., 2017; Wu et al., 2020). As well as, our results demonstrated that warming increased CH4 uptake in the growing season, but decreased CH4 uptake in the non-growing season in the alpine grassland, similar to the results from other grassland ecosystems (Lin et al., 2015; Zhu et al., 2015; Wu et al., 2020). Moreover, a warming-induced soil moisture decrease may be a potential mechanism for the positive influence of warming on soil CH4 uptake in the alpine grassland (Lin et al., 2015). Our results also demonstrated that seasonally asymmetric warming did not significantly affect the response rate of CH4 uptake to temperature increase (Figure 2 d-f, P > 0.05)." L441-502, "Unlike CH4 fluxes in alpine grasslands, Treat et al. (2018) confirmed that nongrowing season wetland were small CH4 sources, and uplands varied from CH4 sinks to CH4 sources." We also revised the sections about "increasing N2O emissions with warming during the non-growing season", and refer to the references you provided. L512-518, "However, our results displayed emissions peaks of N2O during the freeze–thaw periods (e.g., May 2017, June 2018 and April 2019). Warming increased N2O emissions in the thawing period owning to disrupt the gas diffusion barrier and greater C and N availability for microbial activity (Nyborg et al., 1997). Wagner-Riddle et al. (2017) also confirmed that the magnitude of freeze–thaw-induced N2O emissions was related to the number of days with soil temperatures below 0 oC"

Line edits Abstract: L13-14: specify whether CO2 fluxes are ecosystem respiration or net ecosystem exchange, not clear if the reported numbers are net CO2 losses to the atmosphere. Also, do these numbers represent the total range of fluxes between 2016 and 2019? It might be more meaningful to present growing season as well as annual mean or median fluxes in the abstract, as fluxes, especially for CH4 and N2O, are highly variable. Response: Thank you for your precise comment. CO2 fluxes are ecosystem respiration, we defined it exactly in the revised manuscript. See L122 "Ecosystem

respiration (expressed by CO2 flux), N2O, and CH4 fluxes were measured using static chambers..." We agree with this comment, and we revised as "annual mean of CO2, CH4 and N2O fluxes in growing season were 42.83 mg C m-2 h-1, -41.57 $\mu$g C m-2 h-1, and 4.98 $\mu$g N m-2 h-1, respectively." See L13-14.

L14-15: this is counter intuitive and does not match with what is shown in supplementary figure S3 (where CO2 and N2O fluxes show a clear positive correlation with soil temperature, and CH4 uptake increases with increases temperature). Please rephrase. Response: Thank you for your precise comment. We have revised this section as "The CO2 and N2O fluxes were positively correlated with soil temperature, and the CH4 uptake increased with the increase in temperature."

L16-18: "the variation in GHG flux under seasonally asymmetric warming was different between the growing season and the non-growing season": this statement is vague, please be more specific and state clearly what was observed. Response: Thank you for your precise comment. We revised this sentence as "Furthermore, warming in non-growing season increased CO2 and CH4 fluxes in both growing season and non-growing season. However, the increase of N2O flux was mainly caused by the warming of the growing season." See L16-18 of the revised manuscript.

L18-24: a short explanation of the term "response rate" might be needed in this context. Response: We added a sentence to illustrate the term "response rate", the sentence is "the response rate was defined as the change of GHG flux caused by the increase of soil temperature under warming treatment," See L19-20 of the revised manuscript.

L24-27: A clear summary statement with the specific implications of this study would be needed here. Response: we revised the sentence as "We have confirmed the stimulating effect of warming in non-growing season on CO2 and CH4 fluxes, that long-term annual warming weakened the sensitivity of CO2 and N2O fluxes to temperature increase, and that warming in growing season reduced the sensitivity of CO2 fluxes to temperature increase." See L37-41 of the revised manuscript.

Introduction: L38: daytime/nighttime differences are not addressed in this study. Consider removing from the introduction or add results and discussion to address this issue. Response: we removed "and between daytime and nighttime" from the introduction. Relevant references were also deleted. See L52 of the revised manuscript.

L34-36: Yes, but I would advise caution with this statement (considering the larger than average warming in higher latitudes and Arctic amplification). Response: We accepted the comment and removed the sentence. Relevant references were also deleted. See L48 of the revised manuscript.

L39: 3rd assessment report. Response: we revised the term as "3rd". See L50 of the revised manuscript.

L48: delete "in the atmosphere" Response: we deleted "in the atmosphere". See L68 of the revised manuscript.

L47-49: some rephrasing might be needed (considering that water vapour is a major GHG present in Earth's atmosphere as well). Response: we revised the sentence as "Carbon dioxide ($CO_2$), methane ($CH_4$), and nitrous oxide ($N_2O$) are the three of the major greenhouse gases (GHGs) in the atmosphere". See L68-69 of the revised manuscript.

L47-50: a general statement with some background information on the influence of temperature on GHG production and emissions would be useful for the reader in this context. Response: we added the statement as "Experimental warming is known to influence the ecosystem respiration, $CH_4$ uptake and $N_2O$ emission (Parn et al., 2018; Treat et al., 2018; Wang et al., 2019). Information on ecosystem respiration, $CH_4$ uptake and $N_2O$ emission and their sensitivity to warming, will enhance our understanding of ecosystem C and N cycling processes and improve our predictions of the response of ecosystems to global climate change (Li et al., 2020; Wang et al., 2019)." See L71-76 of the revised manuscript.

L50: Please check those % numbers (radiative forcing of CO2 is larger than that of CH4). Response: we checked those % numbers, and revised as ". . .and their contributions to global warming are 60 %, 20 %, and 6 %". See L70 of the revised manuscript.

L50-52: Not sure whether this statement is correct. At least in northern or high-elevation regions with less accessible sites, warming treatments are often conducted during the summer months. Some rephrasing might be needed. Response: we revised this statement as "most studies focus on the influence of warming during the summer months on GHG flux in terrestrial ecosystems". See L77-78 of the revised manuscript.

L55: please specify whether "CO2 flux" in this context refers to increased CO2 emissions or increased net uptake, and under what conditions this increase occurred (warming treatment or naturally warmer summer?). Response: In light of this comment, we have revised this sentence as "A study of the Alaskan tundra found that summer warming (open-top chambers to increase growing season air temperatures) significantly increased the growing season ecosystem respiration by about 20 % (Natali et al., 2011)." See L80-83 of the revised manuscript.

L56-57: for simplicity, replace "the effect of increased temperature in winter" with "the effect of winter warming". Response: we replaced "the effect of increased temperature in winter" with "the slight effect of winter warming". See L83 of the revised manuscript.

L56-58: This sentence is not quite clear. Does this mean winter warming did not affect growing season CO2 fluxes, but winter warming did increase CO2 fluxes during the nongrowing season? Consider rephrasing. Response: we revised the sentence as "Compared with the slight effect of winter warming on the CO2 fluxes in the growing season, warming increased CO2 fluxes during the snow-covered period by more than 50% (Natali et al., 2011)". See L83-85 of the revised manuscript.

L58-60: replace "absorption" with "uptake. Response: we replaced "absorption" with "uptake". See L86 of the revised manuscript.

L62: start sentence with "A study by xx (2012) in an alpine grassland ecosystem showed: : :" Response: Thank you for your accurate comments. We have revised them. See L118-119 of the revised manuscript.

L69: consider added examples for biotic and abiotic factors here. Response: Thank you for your accurate comments. We added the sentence as "For example, over longer time periods of warming, accelerated decomposition and increased plant N uptake may decrease soil organic C and N pools (Wu et al., 2012), and the microbial community with variable C use efficiency may reduce the temperature sensitivity of heterotrophic respiration (Zhou et al., 2011)." See 139-148 of the revised manuscript. Wu, Z., Dijkstra, P., Koch, G. W., & Hungate, B. A. (2012). Biogeochemical and ecological feedbacks in grassland responses to warming. Nature Climate Change, 2, 458–461. https://doi.org/10.1038/nclimate1486 Zhou, J., Xue, K., Xie, J., Deng, Y. e., Wu, L., Cheng, X., . . . Luo, Y. (2011). Microbial mediation of carbon‐cycle feedbacks to climate warming. Nature Climate Change, 2, 106–110. https://doi.org/10.1038/nclimate1331

L75-81: This is rather vague and I suggest to be more specific and clearly state the overall aim of this study. Response: we revised as "Therefore, we hypothesize that warming in non-growing season will stimulate greenhouse gas flux (especially during non-growing season) in alpine steppe. However, continuous warming throughout the year and during the growing season will weaken the sensitivity of greenhouse gas flux to warming. The current short communication will help to evaluate the uncertainty with respect to GHG flux in response to increasing temperatures against the backdrop of global climate change, by carrying out seasonally asymmetrical warming studies in alpine grasslands." See 152-158 of the revised manuscript.

Methods: L87: Is permafrost present at the site? If yes, it would be useful to add this information. Response: Yes, permafrost is present at the site. We added the information as "There was permafrost present at Bayanbulak alpine grassland, and the average maximum frozen depth was more than 250 cm (from 2000 to 2011, Zhang et

al., 2018)." See L164-166 of the revised manuscript.

L93: in addition to soil, vegetation and temperature it would be useful to add some information related to typical soil moisture/water table levels at the site, since that is important for discussing GHG exchange and observed interannual differences. Response: Thank you for your comments, we added the sentence as "and the average annual soil moisture was 5.9 % (2017-2019)". See L182 of the revised manuscript.

L95: please provide description and dimensions of open-top chambers. Response: we added the description and dimensions of open-top chambers as "The open-top chambers (OTCs) were made of 5 mm thick tempered glass. To reduce the impact of precipitation and snow, the OTC with a hexagonal round table which was 100cm high, and the diameters of the bottom and top were 100 cm and 60 cm, respectively." See L183-186 of the revised manuscript.

L101-108: please provide some more details regarding flux measurement and analysis. E.g., were pre-installed collars used for the flux measurement? Were the chambers equipped with a fan and pressure equilibration tube? Were chambers transparent or opaque? Please also mention the sign convention in this context (i.e. positive fluxes = emissions?). Response: we revised these sentences as "Gas samples were taken 0, 10, 20 and 30 minutes after the lid of the static chamber was sealed in between 12:00 and 14:00 (GMT + 8) every day. The rate of ecosystem respiration, $CH_4$ and $N_2O$ fluxes were calculated based on the change in concertation of $CO_2$, $N_2O$ and $CH_4$ in each chamber over time by a linear or non-linear equation ($P < 0.05$, $r^2 > 0.95$) (the positive flux value was emission, and the negative flux value was uptake; Liu et al. 2012; Wang et al. 2013)." See L194-202 of the revised manuscript. Wang K, Zheng X, Pihlatie M, Vesala T, Liu C, Haapanala S, Liu H (2013) Comparison between static chamber and tunable diode laser-based eddy covariance techniques for measuring nitrous oxide fluxes from a cotton field. Agric For Meteorol 171:9–19. Liu C, Wang K, Zheng X (2012) Responses of $N_2O$ and $CH_4$ fluxes to fertilizer nitrogen addition rates in an irrigated wheat-maize cropping system in northern China. Biogeosciences

9:839–850.

L105: please add the total number of sampling times. L105-108: What quality criteria were used to accept or reject fluxes? E.g. r2, RMSE, minimum number of sampling points during one flux measurements? This information is not provided in the cited reference (Chen et al 2013). Response: we added the total number of sampling times as "Gas samples were 232, collecting once or twice a week". And revised L105-108 as "Gas samples were taken 0, 10, 20 and 30 minutes after the lid of the static chamber was sealed in between 12:00 and 14:00 (GMT + 8) every day. The rate of ecosystem respiration, $CH_4$ and $N_2O$ fluxes were calculated based on the change in concertation of $CO_2$, $N_2O$ and $CH_4$ in each chamber over time by a linear or non-linear equation ($P < 0.05$, $R2 > 0.95$) (the positive flux value was emission, and the negative flux value was uptake; Liu et al. 2012; Wang et al. 2013)." We deleted the cited reference (Chen et al 2013) and added the cited references (Liu et al. 2012; Wang et al. 2013). See L192-204 of the revised manuscript.

Results: L124-125: In addition to the range, it would be useful to provide mean/median values here. I also suggest to add a few sentences describing the general pattern of fluxes as this site, before presenting the treatment results, to provide the reader with a general overview (for example stating that the site acted as a net $CH_4$ sink, with negligible $CH_4$ emissions, and small $N_2O$ source). Response: Thank you for your precise comment. We revised the sentences as "Our study showed that the Bayanbulak alpine grassland was a small ecosystem respiration ($CO_2$ flux), a net $CH_4$ sink, and a negligible $N_2O$ source. The annual mean of $CO_2$, $CH_4$ and $N_2O$ fluxes in growing season were 42.83 mg C m-2 h-1, -41.57 $\mu$g C m-2 h-1, and 4.98 $\mu$g N m-2 h-1 from October 2016 to September 2019, respectively (Fig. 1)." See L219-233 the revised manuscript.

L131-136: talking about an increase and decrease in $CH_4$ flux is slightly confusing in this context, as the authors mainly observed $CH_4$ uptake. The 6.4% increase in $CH_4$ flux in the AW treatment that the authors report here is in fact an increase in $CH_4$ uptake, according to Fig. S2. This would mean a decrease in $CH_4$ from an

atmospheric point of view, and I suggest to rephrase this whole section accordingly. To avoid confusion, it might be useful to refer to uptake and emissions, rather than fluxes, throghout the manuscript. Response: Thank you for your precise comment. we rephrased this whole section accordingly. "The AW temperature change induced a 6.4% increase in $CH_4$ uptake in the growing season and a 3.8% decrease in the nongrowing season. The GW treatment resulted in 7.1% and 10.2% increases in $CH_4$ uptake in the growing and nongrowing seasons, respectively. On the contrary, the NGW generated a 10.6% and 9.2 % decrease in $CH_4$ uptake in the growing and nongrowing seasons, respectively (Figure S3 b)." See L240-245 the revised manuscript.

L126-140: are all these reported %changes significant (standard errors seem rather large)? It would be useful to add information regarding statistical significance in Fig S2 and manuscript text. Response: Thank you for your precise comment. One-way ANOVA results of Re, $CH_4$ uptake and $N_2O$ emissions among the four warming treatments were not significant ($P > 0.05$). We added this information in the revised manuscript. See L250-251 of the revised manuscript.

L126 and throughout: please specify which $CO_2$ flux component is discussed (see specific comments above). Response: Thank you for your precise comment. We revised "$CO_2$ flux" as "ecosystem respiration" and abbreviation for Re in the whole manuscript.

L143: delete "extremely". Response: Thank you for your precise comment. We deleted "extremely". See L334 of the revised manuscript.

L141-144: Do the authors mean interannual differences between growing seasons? Response: Yes, Ecosystem respiration (Re), $CH_4$ uptake, and $N_2O$ emissions were distinguished between growing seasons and non-growing seasons in interannual and intertreatment two-way repeated-measure ANOVA. The interannual differences in Re, $CH_4$ uptake, and $N_2O$ emissions were all due to the growing season, except for significant differences in $N_2O$ emissions during the non-growing season. (Figure 1)

[Figure]

L141-145: Generally, this section is not very clear and would benefit from some rephrasing. It would be important to state that interannual differences were larger than impact of warming treatment (for CO2 and N2O) according to Fig. 1, whereas warming treatment had a significant impact on CH4 fluxes. Response: Thank you for your precise comment. We have added the sentences as "Therefore, interannual differences were larger than impact of warming treatment (for Re and N2O emissions) according to Fig. 1, whereas warming treatment had a significant impact on CH4 uptake." in order to better express the meaning of this section. See L381-383 of the revised manuscript.

L147-148: I suggest simplifying "CH4 flux showed significantly decreasing trends with increasing soil temperature" to something like "we observed increasing CH4 uptake with increasing soil temperature". Response: Thank you for your precise comment. We revised as "we observed increasing CH4 uptake with increasing soil temperature", See L385-386 of the revised manuscript.

Discussion and conclusion: Please see my specific comments above regarding the discussion section, as well as: L166-171: It would be useful to include some background information on mechanisms behind CH4 fluxes for the reader, i.e. when do emissions occur, what conditions promote CH4 uptake, why would temperature increase CH4 uptake, etc. Response: Thank you for your precise comment. We revised this section as "Ecosystem CH4 flux is the net result of CH4 production and consumption occur simultaneously under the action of methanogenic archaea and methane-oxidizing bacteria (e.g., Mer and Roger, 2001; Blodau, 2002; Galbally et al., 2008). CH4 fluxes are dependent on temperature, pH, and the availability of substrate (e.g., Moore & Dalva, 1997; Treat et al., 2015). For example, lower levels of soil moisture would decrease C release, indicating that drier soils are the major CH4 sink (Denman et al., 2007). The CH4 uptake observed during the three growing and non-growing seasons implied that the alpine grassland soil could act as an atmospheric CH4 sink, which agrees with the findings of many previous studies in similar region (Li et al., 2015; Wei et al., 2015; Zhao et al., 2017; Wu et al., 2020). As well as, our results demonstrated that warming increased CH4 uptake in the growing season, but decreased CH4 uptake in the non-growing season in the alpine grassland, similar to the results from other grassland ecosystems (Lin et al., 2015; Zhu et al., 2015; Wu et al., 2020). Moreover, a warming-induced soil moisture decrease may be a potential mechanism for the positive influence of warming on soil CH4 uptake in the alpine grassland (Lin et al., 2015). Our results also demonstrated that seasonally asymmetric warming did not significantly affect the response rate of CH4 uptake to temperature increase (Figure 2 d-f, $P > 0.05$). Hu et al. (2016) suggested that asymmetrical responses of CH4 fluxes to warming and cooling should be taken into account when evaluating the effects of climate change on CH4 uptake in the alpine meadow on the Tibetan plateau. The latest research confirmed that warming in the Arctic had been more apparent in the non-growing season than in the typical growing season (Bao et al., 2020). Hereby, Bao et al. (2020) found that the CH4 emissions during spring thaw and autumn freeze contribute to about a quarter of annual total CH4 emissions. That experimental warming is stimulating soil CH4 uptake in the growing season implies that the grasslands of the Bayanbulak may have potential to remove more CH4 from the atmosphere under future global warming conditions." See L421-508 of the revised manuscript.

L176-177: for the nongrowing season contribution of CH4 fluxes a comparison to other ecosystems would be useful; see for example Treat, C.C., Bloom, A.A. and Marushchak, M.E., 2018. Nongrowing season methane emissions–a significant component of annual emissions across northern ecosystems. Global change biology, 24(8), pp.3331-3343. Response: Thank you for your precise comment. We added the sentences as "Unlike CH4 fluxes in alpine grasslands, Treat et al. (2018) confirmed that nongrowing season wetland were small CH4 sources, and uplands varied from CH4 sinks to CH4 sources." See L441-502 of the revised manuscript.

L178-186: please see my specific comment regarding discussion of other environmental variables besides temperature. N2O in particular is rarely depend on just on variable, and the effect of temperature may often be masked by other variables such as

water table, and mineral nitrogen availability. This may require at least short mention in the discussion section. See for example Pärn, J. et al. 2018. Nitrogen-rich organic soils under warm well-drained conditions are global nitrous oxide emission hotspots. Nature communications, 9(1), pp.1-8. Response: Thank you for your precise comment. We added the sentences as "As well as N2O emissions was positively related to soil temperature, Pärn et al. (2018) found that N2O emission from organic soils increases with rising soil NO3-, follows a bell-shaped distribution with soil moisture." See L515-518 of the revised manuscript.

Figures: Fig. 2: please add r2 and P-values for all figure panels (even for nonsignificant relationships). Fig S2: please add number of measurement times for growing season / non-growing season mean. Also, please specify in y-axis or figure caption which component of the CO2 flux is shown (ER?). As panel b shows CH4 uptake, I suggest to flip the y-axis, showing zero on top and negative values at the bottom. Overall, I would suggest to use boxplots (including quartile ranges and outliers) rather than barplots in this figure, to capture the full range of fluxes, as it would be important to show e.g. also the occurrence of emissions (for CH4) or uptake (for N2O). The authors may also consider moving this figure into the main text. Response: Thank you for your precise comment. We added r2 and P-values for all figure panels of the Fig. 2. Based on these comments, we redrew Fig. S2 using boxplots, added number of measurement times for growing season / non-growing season mean. We also moved this figure into the main text.
Interactive
comment

**Different responses of ecosystem respiration, CH₄ uptake, and N₂O emissions to seasonally asymmetric warming in an alpine grassland of Tianshan Mountains**

[Figure]

Figure S2 Variation in soil water content (at 10-cm depth) under four treatments in alpine grassland from June 2017 to September 2019. GS, growing season; NGS, non-growing season; AW, warming throughout the year; NGW, warming in nongrowing season only; GW, warming in growing season only; NW, non-warming. Significant differences among AW, NGW, GW and NW from analysis of variance (ANOVA) are denoted as bars within the same season with different lowercase letters, $P < 0.05$; data points are the mean ± standard error.

**Fig. 1.**

---

## Author Response (AR1)

**A point-by-point response to the reviews**

Dear Reviewers:

Thank you for your comments and the reviewers' comments concerning our manuscript entitled " Different responses of $CO_2$, $CH_4$, and $N_2O$ fluxes to seasonally asymmetric warming in an alpine grassland of Tianshan Mountains" (MS No.: bg-2020-396). Those comments are all valuable and very helpful for revising and improving our paper, as well as the important guiding significance to our researches. We have studied those comments carefully and have made a point to point reply and correction. Revised portion are marked in red color in this manuscript. Specific corrections and responds to the reviewer's comments are listed as follows:

Comments to the Author:

**# the comment by Y.G. Du**

This manuscript describes the response of GHGs emissions to seasonally asymmetric warming in an alpine grassland of Tianshan Mountains. It is an interesting topic to understand carbon and nitrogen cycles with increasing temperature. The manuscript is well written and concise. The experiment is well designed and conducted. I suggest this manuscript could be accepted after some minor revisions.

**Introduction:** the research advances of responses of $CO_2$, $CH_4$ and $N_2O$ fluxes to seasonally asymmetric warming is very limited, more contents could be added especially in grassland ecosystem. Authors quoted many IPCC results about warming and its effect on GHGs fluxes, which need to be summarized.

**Response:** Thank you for your precise comment for the Introduction. We have added to the latest research on the effect of warming on greenhouse gas flux in grassland ecosystems. Add the following: "A recent study showed that seasonal variations in carbon flux were more closely related to air temperature in the meadow steppe (Zhao et al., 2019). Another study found that experimental warming enhanced $CH_4$ uptake in the relatively arid alpine steppe, but had no significant effects on $CH_4$ emission in the moist swamp meadow (Li et al., 2020). Furthermore, soil $CH_4$ uptake was not

significantly affected by warming in the alpine meadow of the Tibetan Plateau (Wu et al., 2020). In contrast, a global meta-analysis showed that experimental warming stimulates C fluxes in grassland ecosystems, and the response of C fluxes to warming strongly varies across the different grassland types, with greater warming responses in cold than in temperate and semi-arid grasslands (Wang et al., 2019). Across the data set, Li et al. (2020) demonstrated that $N_2O$ emissions were significantly enhanced by whole-year warming treatments. In contrast, no significant effects on soil $N_2O$ emissions were observed by in short-season warming." See L321-333 of the revised manuscript.

We also summarized the IPCC results about warming and its effect on GHGs fluxes. The revised content is as follows: "The 3rd and 4rd Assessment Report of the Inter-Governmental Panel on Climate Change (IPCC) proposed that, against the backdrop of global warming, the temperature change shows that the warming amplitude in the winter is greater than that in the summer, with the warming amplitude at high latitude being greater than that at low latitude, and confirmed that the warming shows asymmetric trends on a seasonal scale (Easterling et al., 1997; IPCC, 2001, 2007)." See L124-210 of the revised manuscript.

**Materials and methods:** air temperature and precipitation data of growing season and non-growing season could not be found, which are important to explain the effect of seasonally asymmetric warming on GHGs flux.

Response: Thank you for your comment for the Materials and methods. We have presented air temperature and precipitation data through tow figures. And described the Figure S3 and S4 in the Appendix for the revised manuscript.

[Figure]

Figure S3 Variation in air temperature (inside the open-topped chamber, OTC, 50 cm above the ground) under four treatments in alpine grassland from October 2016 to September 2019. GS, growing season; NGS, non-growing season; AW, warming throughout the year; NGW, warming in non-growing season only; GW, warming in growing season only; NW, non-warming. No significant differences among AW, NGW, GW and NW from analysis of variance (ANOVA) are denoted as bars within the same season with a common lowercase letter, $P < 0.05$; data points are the mean ± standard

[Figure]

error.

Figure S4 Variation in precipitation in the alpine grassland from October 2016 to September 2019. GS, growing season; NGS, non-growing season.

**Discussion:** please delete figure 2 and $P < 0.05$ or $P > 0.05$. The manuscript do not research the response of GHGs to daytime, nighttime or short-season warming, please delete it.

**Response:** Thank you for precise comment for the Discussion. However, we disagree with this comment. Figure 2 shows the highlights of this manuscript: "Response of variations in $CO_2$, $CH_4$, and $N_2O$ fluxes to changes in soil temperature under AW, NGW and GW conditions in the alpine grassland, from 2016 to 2019." Figure 2 does not mention what the comments suggest: "the response of GHGs to daytime, nighttime or short-season warming".

**Conclusions:** please add the responses of $CH_4$ and $N_2O$ fluxes to warming in the study.

**Response:** Thank you for your precise comment for the Conclusions. We have revised the conclusion as "In summary, the effect of seasonally asymmetrical warming on $Re$ and $N_2O$ emission was obvious, unlike the situation with $CH_4$ uptake. The $Re$ and $N_2O$ emission were able to adapt to continuous warming, resulting in a reduced response rates of the $Re$ and $N_2O$ emission to temperature increase. Warming in the non-growing season increased the temperature dependence of the $Re$. Thus, we believe that the study of climate change should pay greater attention to warming in the non-growing season, to avoid underestimating the greenhouse effect on $Re$ in alpine grasslands." See L1090-1096 of the revised manuscript.

**Anonymous Referee #1**

I have reviewed this paper by Gong et al. This study revealed the effect of seasonally asymmetric warming on greenhouse gas fluxes in alpine grassland, and it advances our understanding of warming effects on greenhouse gas fluxes. I think there

are a few minor issues that could be improved Before it could be accepted for publication.

1) The authors should focus more on the mechanisms behind the different responses of greenhouse gas fluxes to seasonally asymmetric warming.

Response: Thank you for your precise comment. We revised the manuscript.

L820-832, "Ecosystem $CH_4$ flux is the net result of $CH_4$ production and consumption, occurring simultaneously under the action of methanogenic archaea and methane-oxidizing bacteria (e.g., Mer and Roger, 2001). In addition, our results demonstrated that warming increased $CH_4$ uptake in the growing season, but decreased $CH_4$ uptake in the non-growing season in the alpine grassland, findings similar to those from other grassland ecosystems (Lin et al., 2015; Wu et al., 2020; Zhu et al., 2015). Our results also demonstrated that seasonally asymmetric warming did not significantly affect the response rate of $CH_4$ uptake (Figure 3 d-f, $P > 0.05$). $CH_4$ flux depended on temperature, pH, and the availability of substrate (e.g., Treat et al., 2015). The $CH_4$ uptake observed during the three growing season and non-growing season implied that the alpine grassland soil could act as an atmospheric $CH_4$ sink, a finding which agrees with the results of many previous studies in similar regions (Wei et al., 2015; Zhao et al., 2017)."

L947-949, "Unlike $CH_4$ flux in alpine grasslands, Treat et al. (2018) confirmed that wetland was a small $CH_4$ source in the non-growing season, whereas uplands varied from $CH_4$ sinks to $CH_4$ sources."

L960-966, "However, our results displayed $N_2O$ emission peaks during the freeze–thaw periods (e.g., May 2017, June 2018 and April 2019). Warming increased $N_2O$ emissions in the thawing period due to disruption of the gas diffusion barrier and greater C and N availability for microbial activity (Nyborg et al., 1997). Wagner-Riddle et al. (2017) also demonstrated that the magnitude of the freeze/thaw-induced $N_2O$ emissions was associated with the number of days with soil temperatures below $0^oC$."

2) There are a few minor things needed to be revised throughout the manuscript.

For example, Line 16: "greenhouse gas flux" should be changed to "greenhouse gas fluxes"; Line 154: "Figure 2" should be "Fig. 2" to be consistent with other places.

Response: Thank you for your precise comment. Line 16: we revised as "greenhouse gas fluxes". See L108 of the revised manuscript.

**Anonymous Referee #2**

General comments

The manuscript "Different responses of CO2, CH4, and N2O fluxes to seasonally asymmetric warming in an alpine grassland of Tianshan Mountains" by Gong et al. describes the effects of seasonal warming (growing season warming, non-growing season warming, annual warming) on CO2, CH4, and N2O fluxes during 3 years at an alpine grassland site in the southern Tianshan mountains, China.

I value the authors efforts to collect multi-year, and year-round data with manual chamber measurements. While I find evaluating the response of GHG fluxes to changes in temperature during different seasons highly important (especially the non-growing season), the manuscript in its current state has several shortcomings, which I have outlined in my specific comments and line edits below.

**Specific comments**

1)  The authors discuss CO2 fluxes, however, it is unclear which component of the CO2 flux has been measured. It would seem that ecosystem respiration was measured, which should be stated clearly throughout the manuscript. Temperature is a well known control on respiration, but based on this study the authors can not draw conclusions on the effect of temperature on net CO2 exchange without accounting for photosynthesis. I suggest revising the manuscript text accordingly.

Response: As the anonymous referee's comment, we're measuring ecosystem respiration. We revised the manuscript text accordingly. See the revised manuscript.

2)  To understand the observed interannual variations in GHG exchange in response to temperature it would be important to include information on other climate parameters as well, especially interannual variations in precipitation, soil moisture and

or water table. If this data is available, I highly recommend including it and to add discussion on this topic.

Response: Thank you for your precise comment. We plot the changes of precipitation, air temperature and soil moisture, and analyze the influence of soil temperature and moisture on greenhouse gas fluxes and their association effect through variance decomposition. we do not have the monitoring data of the groundwater level in Bayinbuluk Grassland. We will carry out the work in the future. It was discussed and analyzed in the revised manuscript. As shown in the figure below:

[Figure]

Figure S2 Variation in soil moisture (at 10-cm depth) under four treatments in alpine grassland from June 2017 to September 2019. GS, growing season; NGS, non-growing season; AW, warming throughout the year; NGW, warming in nongrowing season only; GW, warming in growing season only; NW, non-warming. Significant

differences among AW, NGW, GW and NW from analysis of variance (ANOVA) are denoted as bars within the same season with different lowercase letters, *P* < 0.05; data points are the mean ± standard error.

[Figure]

Figure S3 Variation in air temperature (inside the open-topped chamber, OTC, 50 cm above the ground) under four treatments in alpine grassland from October 2016 to September 2019. GS, growing season; NGS, non-growing season; AW, warming throughout the year; NGW, warming in non-growing season only; GW, warming in growing season only; NW, non-warming. No significant differences among AW, NGW, GW and NW from analysis of variance (ANOVA) are denoted as bars within the same season with a common lowercase letter, *P* < 0.05; data points are the mean ± standard error.

[Figure]

Figure S4 Variation in precipitation in the alpine grassland from October 2016 to September 2019. GS, growing season; NGS, non-growing season.

| | Re | | | CH₄ flux | | | N₂O flux | | |
|---|---|---|---|---|---|---|---|---|---|
| | a | c | b | | | | | | |
| NGW-NGS % | **41.6** | 0.8 | -1.6 | 75.0 | -4.1 | 0.8 | 43.8 | -1.4 | -1.9 |
| NGW-GS % | 6.4 | 6.3 | 9.0 | -2.9 | 0.2 | -2.7 | 1.3 | 4.0 | -0.3 |
| GW-NGS % | 0.7 | **36.5** | **22.2** | 51.3 | 7.4 | 0.9 | **29.6** | 10.2 | -2.0 |
| GW-GS % | **22.6** | -12.4 | **23.4** | -2.6 | 0.4 | -2.4 | 3.8 | 0.9 | <0.1 |
| AW-AY % | 9.5 | **22.3** | 10.1 | 15.3 | 6.2 | -0.9 | 7.7 | 4.5 | -1.9 |
| NW-AY % | 7.6 | **26.7** | 5.0 | 18.5 | 4.7 | -0.9 | **21.5** | -3.7 | 3.5 |

Figure 4 Influence of soil temperature and soil moisture on ecosystem respiration (*Re*), CH₄ uptake, and N₂O emission by variation-partitioning analysis under four treatments in the growing season and non-growing season. a, Single effect of soil temperature (%); b, single effect of soil moisture (%); c, joint effects of soil temperature and moisture (%); NGW-NGS, greenhouse gas fluxes in non-growing season under non-growing season warming treatment; NGW-GS, greenhouse gas fluxes in growing season under non-growing season warming treatment; GW-NGS, greenhouse gas fluxes in non-growing season under growing season warming treatment; GW-GS, greenhouse gas fluxes in growing season under growing season warming treatment; AW-AY, annual

greenhouse gas fluxes under annual warming treatment; NW-AY, annual greenhouse gas fluxes without warming.

3) Adding discussion on whether the warming treatment with OTCs impacted other environmental variables (such as soil moisture, snow depth) would be needed in order to assess the effectiveness of the warming treatment and validity of results. Shortly reporting results on the achieved temperature increase in the warmed plots compared to control treatments in the results section would be helpful as well (this is only shown in the supplementary material Fig S1).

Response: Thank you for your precise comment. Yes, warming treatment with OTCs impacted soil moisture and snow depth. The change of snow depth mainly affects the soil moisture, therefore, we focused on the analysis of how warming affects the greenhouse gas flux by affecting soil moisture (Figure 4). We added the information in the revised manuscript.

4) Introduction as well as discussion and conclusion remain rather superficial. This manuscript would greatly benefit from some streamlining, clearly stating the objectives and relevance of this study, and a more thorough literature review and comparison to other studies. For example, this study reports rather large CH4 uptake rates. How do these rates compare to what is observed in other studies from similar ecosystems? Based on the findings of this study, can larger-scale conclusions be drawn on what impact warming will have on CH4 uptake in these ecosystems? The study also reports all three GHGs, which is a strength of this study, as measurements of N2O fluxes in particular are rare in colder climates. The authors could highlight this in their study, and provide some comparison to other studies. And while investigating the effect of temperature on fluxes was clearly the aim of this study, some acknowledgement of drivers of GHG fluxes other than temperature would be useful to include.

Response: Thank you for your precise comment. We revised the manuscript and added the information of your comments. For example,

L124-210, "The 3rd and 4rd Assessment Report of the Inter-Governmental Panel on Climate Change (IPCC) proposed that, against the backdrop of global warming, the temperature change shows that the warming amplitude in the winter is greater than that in the summer, with the warming amplitude at high latitude being greater than that at low latitude, and confirmed that the warming shows asymmetric trends on a seasonal scale (Easterling et al., 1997; IPCC, 2001, 2007)."

L214-219, "Experimental warming is known to influence ecosystem respiration ($Re$), $CH_4$ uptake, and $N_2O$ emission (Pärn et al., 2018; Treat et al., 2018; Wang et al., 2019). Information on $Re$, $CH_4$ uptake, and $N_2O$ emission and their sensitivity to warming, will enhance our understanding of ecosystem C and N cycling processes and improve our predictions of the response of ecosystems to global climate change (Li et al., 2020; Wang et al., 2019)."

L223-228, "A study of the Alaskan tundra found that summer warming (using open-top chambers to increase air temperatures in the growing season) significantly increased $Re$ in the growing season by about 20 % (Natali et al., 2011). Compared with the slight effect of winter warming on the $CO_2$ fluxes in the growing season, warming increased $CO_2$ fluxes during the snow-covered non-growing season by more than 50% (Natali et al., 2011)."

L321-333, "A recent study showed that seasonal variations in carbon flux were more closely related to air temperature in the meadow steppe (Zhao et al., 2019). Another study found that experimental warming enhanced $CH_4$ uptake in the relatively arid alpine steppe, but had no significant effects on $CH_4$ emission in the moist swamp meadow (Li et al., 2020). Furthermore, soil $CH_4$ uptake was not significantly affected by warming in the alpine meadow of the Tibetan Plateau (Wu et al., 2020). In contrast, a global meta-analysis showed that experimental warming stimulates C fluxes in grassland ecosystems, and the response of C fluxes to warming strongly varies across

the different grassland types, with greater warming responses in cold than in temperate and semi-arid grasslands (Wang et al., 2019). Across the data set, Li et al. (2020) demonstrated that $N_2O$ emissions were significantly enhanced by whole-year warming treatments. In contrast, no significant effects on soil $N_2O$ emissions were observed by in short-season warming."

L398-402, "For example, over longer time periods of warming, accelerated carbon decomposition and increased plant N uptake may decrease soil organic C and N pools (Wu et al., 2012), and the microbial community with variable C use efficiency may reduce the temperature sensitivity of heterotrophic respiration (Zhou et al., 2012)."

L406-409, "Therefore, we hypothesize that warming in the non-growing season will stimulate GHG flux (especially during the non-growing season) in the alpine steppe. However, continuous warming throughout the year and during the growing season will reduce the sensitivity of GHG flux to warming."

L820-832, "Ecosystem $CH_4$ flux is the net result of $CH_4$ production and consumption, occurring simultaneously under the action of methanogenic archaea and methane-oxidizing bacteria (e.g., Mer and Roger, 2001). In addition, our results demonstrated that warming increased $CH_4$ uptake in the growing season, but decreased $CH_4$ uptake in the non-growing season in the alpine grassland, findings similar to those from other grassland ecosystems (Lin et al., 2015; Wu et al., 2020; Zhu et al., 2015). Our results also demonstrated that seasonally asymmetric warming did not significantly affect the response rate of $CH_4$ uptake (Figure 3 d-f, $P > 0.05$). $CH_4$ flux depended on temperature, pH, and the availability of substrate (e.g., Treat et al., 2015). The $CH_4$ uptake observed during the three growing season and non-growing season implied that the alpine grassland soil could act as an atmospheric $CH_4$ sink, a finding which agrees with the results of many previous studies in similar regions (Wei et al., 2015; Zhao et al., 2017)."

L947-949, "Unlike $CH_4$ flux in alpine grasslands, Treat et al. (2018) confirmed

that wetland was a small $CH_4$ source in the non-growing season, whereas uplands varied from $CH_4$ sinks to $CH_4$ sources."

L960-966, "However, our results displayed $N_2O$ emission peaks during the freeze–thaw periods (e.g., May 2017, June 2018 and April 2019). Warming increased $N_2O$ emissions in the thawing period due to disruption of the gas diffusion barrier and greater C and N availability for microbial activity (Nyborg et al., 1997). Wagner-Riddle et al. (2017) also demonstrated that the magnitude of the freeze/thaw-induced $N_2O$ emissions was associated with the number of days with soil temperatures below $0\,^{\circ}C$."

5) The discussion is rather short and exclusively focuses on the reponse rate of the three gases to temperature, while some of the rather interesting key findings of this study are not addressed (such as for example increasing annual CH4 uptake with warming, or increasing N2O emissions with warming during the non-growing season). It also looks like the site displayed emission peaks of N2O during the shoulder periods, especially spring, which might be an important point to mention in the discussion (see for example: Wagner-Riddle et al. (2017). Globally important nitrous oxide emissions from croplands induced by freeze–thaw cycles. Nature Geoscience 10(4):279-83).

Response: Thank you for your precise comment. We revise the sections according to your comments. See the revised manuscript, L820-832, "Ecosystem $CH_4$ flux is the net result of $CH_4$ production and consumption, occurring simultaneously under the action of methanogenic archaea and methane-oxidizing bacteria (e.g., Mer and Roger, 2001). In addition, our results demonstrated that warming increased $CH_4$ uptake in the growing season, but decreased $CH_4$ uptake in the non-growing season in the alpine grassland, findings similar to those from other grassland ecosystems (Lin et al., 2015; Wu et al., 2020; Zhu et al., 2015). Our results also demonstrated that seasonally asymmetric warming did not significantly affect the response rate of $CH_4$ uptake (Figure 3 d-f, $P > 0.05$). $CH_4$ flux depended on temperature, pH, and the availability of substrate (e.g., Treat et al., 2015). The $CH_4$ uptake observed during the three growing season and non-growing season implied that the alpine grassland soil could act as an

atmospheric $CH_4$ sink, a finding which agrees with the results of many previous studies in similar regions (Wei et al., 2015; Zhao et al., 2017)."

L947-949, "Unlike $CH_4$ flux in alpine grasslands, Treat et al. (2018) confirmed that wetland was a small $CH_4$ source in the non-growing season, whereas uplands varied from $CH_4$ sinks to $CH_4$ sources."

We also revised the sections about "Our results suggested that the response of $N_2O$ emission to temperature increase was limited by the warming that occurred throughout the year. However, our results displayed $N_2O$ emission peaks during the freeze–thaw periods (e.g., May 2017, June 2018 and April 2019). Warming increased $N_2O$ emissions in the thawing period due to disruption of the gas diffusion barrier and greater C and N availability for microbial activity (Nyborg et al., 1997). Wagner-Riddle et al. (2017) also demonstrated that the magnitude of the freeze/thaw-induced $N_2O$ emissions was associated with the number of days with soil temperatures below $0^{\circ}C$." See L958-966 of the revised manuscript.

**Line edits**

**Abstract:** L13-14: specify whether CO2 fluxes are ecosystem respiration or net ecosystem exchange, not clear if the reported numbers are net CO2 losses to the atmosphere. Also, do these numbers represent the total range of fluxes between 2016 and 2019? It might be more meaningful to present growing season as well as annual mean or median fluxes in the abstract, as fluxes, especially for CH4 and N2O, are highly variable.

Response: Thank you for your precise comment. CO2 fluxes are ecosystem respiration, we defined it exactly in the revised manuscript. For example, L12 "An experiment was conducted to investigate the effect of seasonally asymmetric warming on ecosystem respiration ($Re$), $CH_4$ uptake, and $N_2O$ emissions in alpine grassland…"

We agree with this comment, and we revised as "…annual mean of $Re$, $CH_4$, and $N_2O$ fluxes in growing season were 42.83 mg C $m^{-2}$ $h^{-1}$, $-41.57$ μg C $m^{-2}$ $h^{-1}$, and 4.98 μg N $m^{-2}$ $h^{-1}$, respectively." See L14-16.

L14-15: this is counter intuitive and does not match with what is shown in supplementary figure S3 (where CO2 and N2O fluxes show a clear positive correlation with soil temperature, and CH4 uptake increases with increases temperature). Please rephrase.

Response: Thank you for your precise comment. We have revised this section as "The *Re*, CH$_4$ uptake, and N$_2$O emissions were positively correlated with soil temperature." See L19-20 of the revised manuscript.

L16-18: "the variation in GHG flux under seasonally asymmetric warming was different between the growing season and the non-growing season": this statement is vague, please be more specific and state clearly what was observed.

Response: Thank you for your precise comment. We revised this sentence as "Furthermore, warming during the non-growing season increased *Re* and CH$_4$ uptake in both the growing season and non-growing seasons. However, the increase in N$_2$O emission in the growing season was mainly caused by the warming during the growing season." See L16-19 of the revised manuscript.

L18-24: a short explanation of the term "response rate" might be needed in this context.

Response: We added a sentence to illustrate the term "response rate", the sentence is "In addition, the response rate was defined by the changes in greenhouse gas fluxes driven by warming." See L22-108 of the revised manuscript.

L24-27: A clear summary statement with the specific implications of this study would be needed here.

Response: we revised the sentence as "we observed the stimulatory effect of warming during the non-growing season on *Re* and CH$_4$ uptake. In contrast, the response rates of *Re* and N$_2$O emissions were gradually attenuated by long-term annual

warming, and the response rate of *Re* was also weakened by warming over the growing season." See L109-112 of the revised manuscript.

**Introduction:** L38: daytime/nighttime differences are not addressed in this study. Consider removing from the introduction or add results and discussion to address this issue.

Response: we removed "and between daytime and nighttime" from the introduction. Relevant references were also deleted.

L34-36: Yes, but I would advise caution with this statement (considering the larger than average warming in higher latitudes and Arctic amplification).

Response: We accepted the comment and removed the sentence. Relevant references were also deleted.

L39: 3rd assessment report.

Response: we revised the term as "3rd". See L124 of the revised manuscript.

L48: delete "in the atmosphere"

Response: we deleted "in the atmosphere".

L47-49: some rephrasing might be needed (considering that water vapour is a major GHG present in Earth's atmosphere as well).

Response: we revised the sentence as "Carbon dioxide ($CO_2$), methane ($CH_4$), and nitrous oxide ($N_2O$) are three of the major greenhouse gases (GHGs) in the atmosphere". See L211-212 of the revised manuscript.

L47-50: a general statement with some background information on the influence of temperature on GHG production and emissions would be useful for the reader in this context.

Response: we added the statement as "Experimental warming is known to influence ecosystem respiration ($Re$), $CH_4$ uptake, and $N_2O$ emission (Pärn et al., 2018; Treat et al., 2018; Wang et al., 2019). Information on $Re$, $CH_4$ uptake, and $N_2O$ emission and their sensitivity to warming, will enhance our understanding of ecosystem C and N cycling processes and improve our predictions of the response of ecosystems to global climate change (Li et al., 2020; Wang et al., 2019)." See L214-219 of the revised manuscript.

L50: Please check those % numbers (radiative forcing of CO2 is larger than that of CH4).

Response: we checked those % numbers, and revised as "with their contributions to global warming being 60 %, 20 %, and 6 %, respectively (IPCC, 2007, 2013).". See L213 of the revised manuscript.

L50-52: Not sure whether this statement is correct. At least in northern or high-elevation regions with less accessible sites, warming treatments are often conducted during the summer months. Some rephrasing might be needed.

Response: we revised this statement as "At present, most studies focus on the influence of warming on GHG flux in terrestrial ecosystems during the summer months (Keenan et al., 2014; Li et al., 2011; Yang et al., 2014).". See L220-222 of the revised manuscript.

L55: please specify whether "CO2 flux" in this context refers to increased CO2 emissions or increased net uptake, and under what conditions this increase occurred (warming treatment or naturally warmer summer?).

Response: In light of this comment, we have revised this sentence as "A study of the Alaskan tundra found that summer warming (using open-top chambers to increase air temperatures in the growing season) significantly increased $Re$ in the growing season by about 20 % (Natali et al., 2011)." See L223-226 of the revised manuscript.

L56-57: for simplicity, replace "the effect of increased temperature in winter" with "the effect of winter warming".

Response: we replaced "the effect of increased temperature in winter" with "the slight effect of winter warming". See L226 of the revised manuscript.

L56-58: This sentence is not quite clear. Does this mean winter warming did not affect growing season CO2 fluxes, but winter warming did increase CO2 fluxes during the nongrowing season? Consider rephrasing.

Response: we revised the sentence as "Compared with the slight effect of winter warming on the $CO_2$ fluxes in the growing season, warming increased $CO_2$ fluxes during the snow-covered non-growing season by more than 50% (Natali et al., 2011)." See L226-228 of the revised manuscript.

L58-60: replace "absorption" with "uptake.

Response: we replaced "absorption" with "uptake". See L229 of the revised manuscript.

L62: start sentence with "A study by xx (2012) in an alpine grassland ecosystem showed: : :"

Response: Thank you for your accurate comments. We have revised them. See L316-317 of the revised manuscript.

L69: consider added examples for biotic and abiotic factors here.

Response: Thank you for your accurate comments. We added the sentence as "For example, over longer time periods of warming, accelerated carbon decomposition and increased plant N uptake may decrease soil organic C and N pools (Wu et al., 2012), and the microbial community with variable C use efficiency may reduce the temperature sensitivity of heterotrophic respiration (Zhou et al., 2012)." See 398-402

of the revised manuscript.

Wu, Z., Dijkstra, P., Koch, G. W., & Hungate, B. A. (2012). Biogeochemical and ecological feedbacks in grassland responses to warming. Nature Climate Change, 2, 458–461. https://doi.org/10.1038/nclimate1486

Zhou, J., Xue, K., Xie, J., Deng, Y. e., Wu, L., Cheng, X., … Luo, Y. (2012). Microbial mediation of carbon-cycle feedbacks to climate warming. Nature Climate Change, 2, 106–110. https://doi.org/10.1038/nclimate1331

L75-81: This is rather vague and I suggest to be more specific and clearly state the overall aim of this study.

Response: we revised as "Therefore, we hypothesize that warming in the non-growing season will stimulate GHG flux (especially during the non-growing season) in the alpine steppe. However, continuous warming throughout the year and during the growing season will reduce the sensitivity of GHG flux to warming. This current short communication will help to assess this variation with respect to GHG flux response to increasing temperatures against the backdrop of global climate change, by carrying out seasonally asymmetrical warming studies in alpine grasslands." See 406-412 of the revised manuscript.

**Methods:** L87: Is permafrost present at the site? If yes, it would be useful to add this information.

Response: Yes, permafrost is present at the site. We added the information as "Permafrost is present in the Bayinbuluk alpine grassland, with the average maximum frozen depth (from 2000 to 2011, Zhang et al., 2018) being more than 250 cm." See L418-443 of the revised manuscript.

L93: in addition to soil, vegetation and temperature it would be useful to add some information related to typical soil moisture/water table levels at the site, since that is important for discussing GHG exchange and observed interannual differences.

Response: Thank you for your comments, we added the sentence as "and the average annual soil moisture was 5.9 % (2017-2019)". See L451 of the revised

manuscript.

L95: please provide description and dimensions of open-top chambers.

Response: we added the description and dimensions of open-top chambers as "The open-top chambers (OTCs) were made of 5 mm thick tempered glass. To reduce the impact of precipitation and snow, the OTC was constructed with a hexagonal round table which was 100 cm high, and the diagonals of the bottom and top were 100 cm and 60 cm, respectively." See L452-455 of the revised manuscript.

L101-108: please provide some more details regarding flux measurement and analysis. E.g., were pre-installed collars used for the flux measurement? Were the chambers equipped with a fan and pressure equilibration tube? Were chambers transparent or opaque? Please also mention the sign convention in this context (i.e. positive fluxes = emissions?).

Response: we revised these sentences as "Gas samples were taken 0, 10, 20 and 30 minutes after the lid of the static chamber was sealed in between 12:00 and 14:00 (GMT + 8) every day. The rates of ecosystem respiration, $CH_4$ and $N_2O$ fluxes were calculated based on the change in concentration of $CO_2$, $N_2O$ and $CH_4$ in each chamber over time by a linear or non-linear equation ($P < 0.05$, $r^2 > 0.95$) (the positive flux values represent emission, and the negative flux values represent uptake; Liu et al. 2012; Wang et al. 2013)." See L508-513 of the revised manuscript.

Wang K, Zheng X, Pihlatie M, Vesala T, Liu C, Haapanala S, Liu H (2013) Comparison between static chamber and tunable diode laser-based eddy covariance techniques for measuring nitrous oxide fluxes from a cotton field. Agric For Meteorol 171:9–19.

Liu C, Wang K, Zheng X (2012) Responses of N2O and CH4 fluxes to fertilizer nitrogen addition rates in an irrigated wheat-maize cropping system in northern China. Biogeosciences 9:839–850.

L105: please add the total number of sampling times. L105-108: What quality criteria were used to accept or reject fluxes? E.g. r2, RMSE, minimum number of

sampling points during one flux measurements? This information is not provided in the cited reference (Chen et al 2013).

Response: we added the total number of sampling times as "A total of 232 samples were taken, collecting once or twice a week." And revised L105-108 as "Gas samples were taken 0, 10, 20 and 30 minutes after the lid of the static chamber was sealed in between 12:00 and 14:00 (GMT + 8) every day. The rates of ecosystem respiration, $CH_4$ and $N_2O$ fluxes were calculated based on the change in concentration of $CO_2$, $N_2O$ and $CH_4$ in each chamber over time by a linear or non-linear equation ($P < 0.05$, $r^2 > 0.95$) (the positive flux values represent emission, and the negative flux values represent uptake; Liu et al. 2012; Wang et al. 2013).. See L508-513 of the revised manuscript.

**Results:** L124-125: In addition to the range, it would be useful to provide mean/median values here. I also suggest to add a few sentences describing the general pattern of fluxes as this site, before presenting the treatment results, to provide the reader with a general overview (for example stating that the site acted as a net CH4 sink, with negligible CH4 emissions, and small N2O source).

Response: Thank you for your precise comment. We revised the sentences as "Our study showed that the Bayinbuluk alpine grassland exhibited a low $Re$, was a net $CH_4$ sink, and a negligible $N_2O$ source. The annual mean values of $Re$, $CH_4$ uptake, and $N_2O$ emissions in the growing season were 42.83 mg C $m^{-2}$ $h^{-1}$, 41.57 μg C $m^{-2}$ $h^{-1}$, and 4.98 μg N $m^{-2}$ $h^{-1}$ , respectively, from October 2016 to September 2019 (Figure 1)." See L570-573 the revised manuscript.

L131-136: talking about an increase and decrease in CH4 flux is slightly confusing in this context, as the authors mainly observed CH4 uptake. The 6.4% increase in CH4 flux in the AW treatment that the authors report here is in fact an increase in CH4 uptake, according to Fig. S2. This would mean a decrease in CH4 from an atmospheric point of view, and I suggest to rephrase this whole section accordingly. To avoid confusion, it might be useful to refer to uptake and emissions, rather than fluxes, throughout the

manuscript.

Response: Thank you for your precise comment. we rephrased this whole section accordingly. "The AW temperature change induced a 6.4% increase in $CH_4$ uptake in the growing season and a 3.8% decrease in the non-growing season. The GW treatment resulted in 7.1% and 10.2% increases in $CH_4$ uptake in the growing season and non-growing season, respectively. On the contrary, the NGW generated a 10.6% and 9.2 % decrease in $CH_4$ uptake in the growing season and non-growing season, respectively (Figure 2 b)." See L579-584 the revised manuscript.

L126-140: are all these reported % changes significant (standard errors seem rather large)? It would be useful to add information regarding statistical significance in Fig S2 and manuscript text.

Response: Thank you for your precise comment. One-way ANOVA results of $Re$, $CH_4$ uptake and $N_2O$ emissions among the four warming treatments were not significant ($P > 0.05$). We added this information in the revised manuscript, "One-way ANOVA results of $Re$, $CH_4$ uptake and $N_2O$ emissions among the four warming treatments were not significant, with the exception that the soil $CH_4$ uptake in the growing season 2019 under GW treatment was significantly higher than that of the AW and NGW treatments ($P < 0.05$).". See L589-697 of the revised manuscript.

L126 and throughout: please specify which CO2 flux component is discussed (see specific comments above).

Response: Thank you for your precise comment. We revised "$CO_2$ flux" as "ecosystem respiration" and abbreviation for $Re$ in the whole manuscript.

L143: delete "extremely".

Response: Thank you for your precise comment. We deleted "extremely".

L141-144: Do the authors mean interannual differences between growing seasons?

Response: Yes, Ecosystem respiration (*Re*), CH$_4$ uptake, and N$_2$O emissions were distinguished between growing seasons and non-growing seasons in interannual and intertreatment two-way repeated-measure ANOVA. The interannual differences in *Re*, CH$_4$ uptake, and N$_2$O emissions were all due to the growing season, except for significant differences in N$_2$O emissions during the non-growing season. (Figure 1)

L141-145: Generally, this section is not very clear and would benefit from some rephrasing. It would be important to state that interannual differences were larger than impact of warming treatment (for CO2 and N2O) according to Fig. 1, whereas warming treatment had a significant impact on CH4 fluxes.

Response: Thank you for your precise comment. We have added the sentences as "Therefore, interannual variation was larger than the impact of the warming treatment (for *Re* and N$_2$O emissions, Figure 1), whereas the warming treatment had a significant impact on CH$_4$ uptake." in order to better express the meaning of this section. See L703-705 of the revised manuscript.

L147-148: I suggest simplifying "CH4 flux showed significantly decreasing trends with increasing soil temperature" to something like "we observed increasing CH4 uptake with increasing soil temperature".

Response: Thank you for your precise comment. We revised as "we observed increasing CH4 uptake with increasing soil temperature", See L707 of the revised manuscript.

**Discussion and conclusion:** Please see my specific comments above regarding the discussion section, as well as:

L166-171: It would be useful to include some background information on mechanisms behind CH4 fluxes for the reader, i.e. when do emissions occur, what conditions promote CH4 uptake, why would temperature increase CH4 uptake, etc.

Response: Thank you for your precise comment. We revised this section as

"Ecosystem $CH_4$ flux is the net result of $CH_4$ production and consumption, occurring simultaneously under the action of methanogenic archaea and methane-oxidizing bacteria (e.g., Mer and Roger, 2001). In addition, our results demonstrated that warming increased $CH_4$ uptake in the growing season, but decreased $CH_4$ uptake in the non-growing season in the alpine grassland, findings similar to those from other grassland ecosystems (Lin et al., 2015; Wu et al., 2020; Zhu et al., 2015). Our results also demonstrated that seasonally asymmetric warming did not significantly affect the response rate of $CH_4$ uptake (Figure 3 d-f, $P > 0.05$). $CH_4$ flux depended on temperature, pH, and the availability of substrate (e.g., Treat et al., 2015). The $CH_4$ uptake observed during the three growing season and non-growing season implied that the alpine grassland soil could act as an atmospheric $CH_4$ sink, a finding which agrees with the results of many previous studies in similar regions (Wei et al., 2015; Zhao et al., 2017). Hu et al. (2016) suggested that asymmetrical responses of $CH_4$ fluxes to warming and cooling should be taken into account when evaluating the effects of climate change on $CH_4$ uptake in the alpine meadow on the Tibetan plateau. Unlike $CH_4$ flux in alpine grasslands, Treat et al. (2018) confirmed that wetland was a small $CH_4$ source in the non-growing season, whereas uplands varied from $CH_4$ sinks to $CH_4$ sources. The latest research confirmed that warming in the Arctic had become more apparent in the non-growing season than in the typical growing season (Bao et al., 2020). Hereby, Bao et al. (2020) found that the $CH_4$ emissions during the spring thaw and the autumn freeze contributed approximately one-quarter of the annual total $CH_4$ emissions. That experimental warming is stimulating soil $CH_4$ uptake in the growing season implies that the grasslands of the Bayinbuluk may have the potential to remove more $CH_4$ from the atmosphere under future global warming conditions." See L822-958 of the revised manuscript.

L176-177: for the nongrowing season contribution of CH4 fluxes a comparison to other ecosystems would be useful; see for example Treat, C.C., Bloom, A.A. and Marushchak, M.E., 2018. Nongrowing season methane emissions–a significant component of annual emissions across northern ecosystems. Global change biology, 24(8), pp.3331-3343.

Response: Thank you for your precise comment. We added the sentences as "Unlike $CH_4$ flux in alpine grasslands, Treat et al. (2018) confirmed that wetland was a small $CH_4$ source in the non-growing season, whereas uplands varied from $CH_4$ sinks to $CH_4$ sources." See L949-951 of the revised manuscript.

L178-186: please see my specific comment regarding discussion of other environmental variables besides temperature. N2O in particular is rarely depend on just on variable, and the effect of temperature may often be masked by other variables such as water table, and mineral nitrogen availability. This may require at least short mention in the discussion section. See for example Pärn, J. et al. 2018. Nitrogen-rich organic soils under warm well-drained conditions are global nitrous oxide emission hotspots. Nature communications, 9(1), pp.1-8.

Response: Thank you for your precise comment. We added the sentences as "Pärn et al. (2018) found that $N_2O$ emission from organic soils increases with rising soil $NO_3^-$, follows a bell-shaped distribution with soil moisture." See L969-971 of the revised manuscript.

**Figures:** Fig. 2: please add r2 and P-values for all figure panels (even for nonsignificant relationships). Fig S2: please add number of measurement times for growing season / non-growing season mean. Also, please specify in y-axis or figure caption which component of the CO2 flux is shown (ER?). As panel b shows CH4 uptake, I suggest to flip the y-axis, showing zero on top and negative values at the bottom. Overall, I would suggest to use boxplots (including quartile ranges and outliers)

rather than barplots in this figure, to capture the full range of fluxes, as it would be important to show e.g. also the occurrence of emissions (for CH4) or uptake (for N2O). The authors may also consider moving this figure into the main text.

Response: Thank you for your precise comment. We added $r^2$ and $P$-values for all figure panels of the Fig. 3 in the revised manuscript. Based on these comments, we redrew Fig. S2 using boxplots (Now switch to Figure 2 in the revised manuscript), added number of measurement times for growing season / non-growing season mean.

---

## Referee Report (RR1)

**General comments**

The manuscript titled "Different responses of ecosystem respiration, $CH_4$ uptake, and $N_2O$ emissions to seasonally asymmetric warming in an alpine grassland of the Tianshan Mountains" by Gong et al. talks about the responses of the three GHG fluxes viz., $CO_2$ (ecosystem $CO_2$ efflux), $CH_4$ and $N_2O$ to different seasonal (growing and non-growing season) and annual experimental warming across 3 years in an alpine grassland on southern Tianshan mountain.

The manuscript investigates an important question with a strategic experimental design and intensive data collection. Most questions raised so far have been answered and revisions made are acceptable. In spite of this the manuscript in the current state has significant drawbacks due lack of information at some places. At other places information is given without providing the context for the same.

The manuscript requires further revision and the general concerns are given below:

- Lack of information in methodology regarding OTC installation strategy, selection of sampling time of GHG fluxes, microclimatic parameters measured and few data analyses.

- The discussion does not flow from results and at places results are written in discussion section (example line no. 251-260). The discussion section focuses on response rates (RR) of GHG fluxes but the results section does not mention RR.

- It is suggested to compute $Q_{10}$ which is a direct and widely used parameter to assess temperature sensitivity (see Zhou et al., 2016 "Experimental warming of a mountain tundra increases soil $CO_2$ effluxes and enhances $CH_4$ and $N_2O$ uptake at Changbai Mountain, China").

- The magnitude of temperature increase (both air and soil) inside open-top chambers should be mentioned. The study is based on the premise of significant warming within the OTC; however, figures indicate otherwise. The air temperature did not significantly increase during non-growing season in any of the plots (GW, NGW and AW) whereas the soil temperature did not significantly increase in any of the season (both growing and non-growing) and plots in the entire 3-year study. Though non-significant, there is an increasing trend in temperatures inside OTCs which should be discussed.

- Findings indicate strong influence of moisture on the GHG fluxes and should be discussed. Both $R_e$ (during growing season) and $N_2O$ uptake varied interannually, coinciding with the variations in moisture. The study area is comparatively drier in

comparison to other alpine grasslands of the world hence moisture is likely to be a limiting factor. Moisture reduction inside OTCs can have significant influence on microbial enzyme activities and eventually on uptake and emission of GHGs.

**Section wise comments for major revision**

**Introduction**

1. The hypothesis does not directly relate to the objectives or the results of the work as the study focuses on seasonally asymmetric warming and continuous measurement of $Re$ and $CH_4$-$N_2O$ fluxes over 3-years.

**Methodology**

2. When were the OTCs installed or removed? Please clarify? For example, "for continuous annual warming OTCs remained installed since the beginning of the study while for growing season warming, these were installed at the onset of growing season and removed at the end of growing season…."

3. Why was the sampling performed only between 12:00 and 14:00 (GMT + 8) every day (line no. 133-134)? Was this time standardisation based on time interval coinciding with mean of diurnal (over 24 hrs) flux rates?

4. Line no. 116 states that all the plots were ungrazed since 2005, how was this achieved? I assume the plots or the entire site was fenced. Please clarify?

5. Measurement of soil temperature and soil moisture at 10 cm depth by data loggers were made at what frequency? hourly or daily? How air temperature was measured or recorded inside all the 4 experimental plots and at what height?

6. Line no. 134 states that the gas samples were collected every day while line no. 139, in contrast, states that they were collected once or twice a week. Clarify.

7. Line no. 138-139 states that "A total of 232 samples were taken, collecting once or twice a week" however figure 2 shows that n = 232 only for the growing season of 2017 whereas n= 192 for GS 2018 and n= 128 for others. Kindly correct.

8. One-way ANOVA was performed to compare only soil temperature (line no. 144). As Figures S2 and S3 indicate that you performed ANOVA for soil moisture and air temperature also, correct your statement.

9. General linear analysis was carried out between soil temperature and GHG fluxes only. The same analysis could be repeated for soil moisture also.

10. Use of variation partitioning analysis in figure 4 should be mentioned under methodology.

**Results**

11. Results of all the GHG fluxes under warming have been given in terms of increase or decrease however, the ANOVA results do not show significant difference, which should be mentioned and discussed. For example, in line no. 160-161: "Compared with the control group (NW), the *Re* was decreased by 7.5% and 4.0% in the growing season and non-growing season, respectively, under AW" add "however non-significant" Alternatively, write line no. 175-179 (stating ANOVA results) before line no. 160, so as to report in the beginning only, that the differences were not significant.

12. Line no. 172: increase in $N_2O$ emission by 101.9% and 192.3% under AW and NGW in the growing season seems very high. Please check.

13. The authors may fit an exponential curve to determine the relationship between Re and soil temperature at 10 cm depth. Figure S5 a indicate towards an exponential pattern.

**Discussion**

14. Results should report response rates (only given in discussion section). The low $r^2$ value of linear regression in Figure 3 (where significant) should be discussed.

15. Line no. 251-260 merely gives results of variation partitioning analysis without any interpretation. This analysis should be mentioned in methodology and in result section.

**Minor edits**

Line 1: Although it is known that ecosystem respiration means $CO_2$ emissions, why not specify $CO_2$ instead for coining "respiration" as done for other gases (as done in the previous draft). This will avoid ambiguity in title. for example,

"Different responses of ecosystem $CO_2$ and $N_2O$ emissions and $CH_4$ uptake to seasonally asymmetric warming in an alpine grassland of the Tianshan Mountains"

Also delete comma after $CH_4$ uptake.

Line 16-19: specify percentage increase for each GHG flux.

Line no. 26: remove comma after annual warming.

Line no. 31: (i) Write greenhouse gas fluxes instead of flux, (ii) as the manuscript doesn't include temperature sensitivity as objective and it has not been calculated, it is not logical to use it as a keyword.

Line no. 35-37: Shorten the sentence as "The global surface temperature increased by about 0.85°C from 1880 to 2012 and is expected to increase by about 1.1–6.4°C by the end of this century (IPCC, 2007, 2013).

Line no. 38: remove comma after scale.

Line no. 52: The warming or the temperature sensitivity of the GFG fluxes have not been evaluated in the study and hence "and their sensitivity to warming" may be removed.

Line no. 60. Remove space between numeric and percentage sign. Follow this in the entire manuscript.

Line no. 62: what do you mean by $CO_2$ fluxes? Is it respiration (if yes is it soil or ecosystem) or photosynthesis or both? Consider this in Line no. 78 also.

Line no. 63-67: as you are stating the result of specific study (Lin et al., 2015), it is better to start the sentence as "Lin et al. (2015) reported….". Also give the percentage increase in $CH_4$ uptake under growing season also. Alternatively, you may add more references of the studies showing similar results, in this the percentage increase may be removed.

Lime no. 85: replace GHG flux with GHG fluxes. Follow this in Line no. 88, 98, 100.

Line no. 111: add space between $-4.8$ and °C.

Line no. 157-159: Figure 1 does not show the annual mean values of each flux but the variations during each year and hence the reference to figure 1 in this sentence is redundant.

**Suggestions**

1. Microclimatic parameters such as air and soil temperature and soil moisture are important to understand variations in seasonal, inter-annual and the asymmetric warming effect on GHG fluxes. Hence these should be included in the main text and their methodology and results should be stated and used while interpreting warming or inter-annual effects on GHG fluxes.

2. Calculate $Q_{10}$ values (atleast for ecosystem respiration). This will give you a more direct indication of temperature sensitivity changes with warming. As most studies use this approach, it will be useful for comparison.

3. In Figure 2, it is suggested to add boxes for mean (entire study period) of each flux rates during growing and non-growing season under four treatments along with ANOVA results (as letters).

---

## Referee Report (RR2)

The manuscript titled "Different responses of ecosystem $CO_2$ and $N_2O$ emissions and $CH_4$ uptake to seasonally asymmetric warming in an alpine grassland of the Tianshan Mountains" by Gong et al. may be accepted after the following minor edits:

1. The authors, in this revision, state that they measured humidity (line no. 136). The variations in humidity (I assume relative humidity) under warming as given for other parameters should be provided in supplementary. Alternatively, I suggest that humidity may be removed as it is not discussed further in the manuscript.

2. In line no. 138 correct "OCTs" to "OTCs"

3. In figure 3, If you provide correlation (r value) instead of regression ($r^2$ value) then in line no. 476 (figure 3) correct "presented by linear regression" to "presented by linear correlation".

4. My earlier suggestion regarding figure 2 was to add **additional boxes** to represent overall variations in GHG fluxes (entire sampling period from 2016-2019). It will read as (in X-axis):

   GS 2017       GS 2018       GS 2019       GS Mean

   Do the same for NGS. Mean lines (shown in red) are not required.

---

## Author Response (AR2)

**A point-by-point reply**

Dear Associate Editor, Anonymous Referee #1 and #3:

Thank you for your comments and the reviewers' comments concerning our manuscript entitled " Different responses of ecosystem respiration, $CH_4$ uptake, and $N_2O$ emissions to seasonally asymmetric warming in an alpine grassland of the Tianshan Mountains " (MS No.: bg-2020-396). Those comments are all valuable and very helpful for revising and improving our paper, as well as the important guiding significance to our researches. We have studied those comments carefully and have made a point to point reply and correction. Revised portion are marked in red color in this manuscript. Specific corrections and responds to the Referee's comments are listed as follows:

**General comments**

The manuscript titled "Different responses of ecosystem respiration, $CH_4$ uptake, and $N_2O$ emissions to seasonally asymmetric warming in an alpine grassland of the Tianshan Mountains" by Gong et al. talks about the responses of the three GHG fluxes viz., $CO_2$ (ecosystem $CO_2$ efflux), $CH_4$ and $N_2O$ to different seasonal (growing and non-growing season) and annual experimental warming across 3 years in an alpine grassland on southern Tianshan mountain.

The manuscript investigates an important question with a strategic experimental design and intensive data collection. Most questions raised so far have been answered and revisions made are acceptable. In spite of this the manuscript in the current state has significant drawbacks due lack of information at some places. At other places information is given without providing the context for the same.

The manuscript requires further revision and the general concerns are given below:

- Lack of information in methodology regarding OTC installation strategy, selection of sampling time of GHG fluxes, microclimatic parameters measured and few data analyses.

**Response:** Thank you for your precise comment. We supplemented the information in methodology regarding OTC installation strategy, see line no. 126 and 127 "After the warming in the NGW or GW, the tempered glass was removed and the frame was retained." Samples were taken once or twice a week, See line no. 143. Microclimate parameters were provided in the appendix, as shown in Figure S1-S3. "Soil temperature and air temperature were increased about 2.3 $^oC$ and 4 $^oC$ by the warming treatment, respectively (Figure S1 and S3). Soil moisture was reduced about 5 % by the warming treatment (Figure S2)." See line no. 132-134.

- The discussion does not flow from results and at places results are written in discussion section (example line no. 251-260). The discussion section focuses on response rates (RR) of GHG fluxes but the results section does not mention RR.

**Response:** Thank you for your precise comment. This section (line no. 251-260) has been moved to the results, see line no. 197-208. Response rates (RR) of GHG fluxes are mainly used to reveal the effect of temperature change on GHG flux. It is more valuable to clarify the relationship between temperature change and the GHG flux change in the Discussion. In response to the previous comment, we simply analyzed the microclimate variation in our experimental method, so we did not mention the RR value of seasonal asymmetric warming in our results.

- It is suggested to compute $Q_{10}$ which is a direct and widely used parameter to assess temperature sensitivity (see Zhou et al., 2016 "Experimental warming of a mountain tundra increases soil $CO_2$ effluxes and enhances $CH_4$ and $N_2O$ uptake at Changbai Mountain, China").

**Response:** Thank you for your precise comment. In future research work, we will pay more attention to the calculation of $Q_{10}$ to evaluate the temperature sensitivity of greenhouse gas flux. Thanks again.

- The magnitude of temperature increase (both air and soil) inside open-top chambers should be mentioned. The study is based on the premise of significant warming within the OTC; however, figures indicate otherwise. The air temperature did not significantly increase during non-growing season in any of the plots (GW, NGW and AW) whereas the soil temperature did not significantly increase in any of the season (both growing and non-growing) and plots in the entire 3-year study. Though non-significant, there is an increasing trend in temperatures inside OTCs which should be discussed.

**Response:** Thank you for your precise comment. Through three years of experimental monitoring, it is found that open-top warming significantly increases the air temperature in growing season (P<0.05, Figure S3 a); The air temperature increases in non-growing season, but not significantly, and the temperature increase range is about 4 ºC in general. Compared with Non-warming, annual warming, warming in non-growing season only, and warming in growing season only all achieved warming effects (Figure S3 b). Similarly, soil temperature was changed by different warming treatments, but the increase rate was lower than that of air temperature. In both the growing season and non-growing season, the increase rate was between 1.5 ºC - 3 ºC, with no significant change. However, the warming effect required by the study was

achieved (Figure S1).

- Findings indicate strong influence of moisture on the GHG fluxes and should be discussed. Both $R_e$ (during growing season) and $N_2O$ uptake varied interannually, coinciding with the variations in moisture. The study area is comparatively drier in comparison to other alpine grasslands of the world hence moisture is likely to be a limiting factor. Moisture reduction inside OTCs can have significant influence on microbial enzyme activities and eventually on uptake and emission of GHGs.

Response: Thank you for your precise comment. General linear analysis was carried out between soil moisture and GHG fluxes, Figure S6 was drawn and added to the Appendix. And the relevant sentences are added as "General linear analyses were used to identify significant linear correlations and regressions between soil temperature and moisture variations and the responses of $Re$, $CH_4$ uptake, or $N_2O$ emissions, respectively." See line no. 164-170.
"However, $Re$, $CH_4$ uptake and $N_2O$ emission were no significant linearly correlated with soil moisture, respectively ($P \geq 0.05$; Figure S6)." See line no. 217, 218.
In future studies, we will focus on the effect of soil moisture reduction caused by warming on microbial enzyme activities and the consequent change of greenhouse gas fluxes.

**Section wise comments for major revision**

**Introduction**

1. The hypothesis does not directly relate to the objectives or the results of the work as the study focuses on seasonally asymmetric warming and continuous measurement of $Re$ and $CH_4$-$N_2O$ fluxes over 3-years.

Response: Thank you for your precise comment for the Introduction. We have revised this sentence as "we hypothesize the stimulatory effect of warming during the non-growing season on $Re$, $CH_4$ uptake and $N_2O$ emissions, the response rates of $Re$, $CH_4$ uptake and $N_2O$ emissions were gradually attenuated by long-term annual warming and warming over the growing season, respectively." See line no. 97-100.

**Methodology**

2. When were the OTCs installed or removed? Please clarify? For example, "for continuous annual warming OTCs remained installed since the beginning of the study while for growing season warming, these were installed at the onset of growing season

and removed at the end of growing season…."

Response: Thank you for your precise comment. For the installation and removal of OTCs, we briefly explained in the Methodology, see line no. 132-136.

3. Why was the sampling performed only between 12:00 and 14:00 (GMT + 8) every day (line no. 133-134)? Was this time standardisation based on time interval coinciding with mean of diurnal (over 24 hrs) flux rates?

Response: Thank you for your precise comment. Yes, the sampling performed between 12:00 and 14:00 (GMT + 8) every day which based on time interval coinciding with mean of diurnal (over 24 hrs) flux rates. The diurnal variation of greenhouse gases flux rates is shown in the figures below:

[Figure]

Figure R1 Diurnal variation of $CO_2$ flux in the non-growing season (NGS) and growing season (GS). AW, warming throughout the year; NGW, warming in the non-growing season only; GW, warming in the growing season only; NW, non-warming.

[Figure]

Figure R2 Diurnal variation of $CH_4$ flux in the non-growing season (NGS) and growing season (GS). AW, warming throughout the year; NGW, warming in the non-growing season only; GW, warming in the growing season only; NW, non-warming.

[Figure]

Figure R3 Diurnal variation of $N_2O$ flux in the growing season (GS). AW, warming throughout the year; NGW, warming in the non-growing season only; GW, warming in the growing season only; NW, non-warming.

4.  Line no. 116 states that all the plots were ungrazed since 2005, how was this achieved? I assume the plots or the entire site was fenced. Please clarify?

Response: Thank you for your precise comment. We revised the sentence as "The site was fenced since 2005,…" See line no. 125.

5.  Measurement of soil temperature and soil moisture at 10 cm depth by data loggers were made at what frequency? hourly or daily? How air temperature was measured or recorded inside all the 4 experimental plots and at what height?

Response: Thank you for your precise comment. Soil temperature and moisture (10cm) were measured at a frequency of every half an hour. The air temperature and humidity inside the OTCs is also recorded at a frequency of every half an hour using HOBO Pro RH/TEMP Data LOGGERS (hanged in the center of the OCTs, 50cm above the surface). We revised this sentence. See line no. 138-146.

6.  Line no. 134 states that the gas samples were collected every day while line no. 139, in contrast, states that they were collected once or twice a week. Clarify.

Response: Thank you for your precise comment. We revised this sentence as "…, collecting once or twice a week." See line no. 153,154.

7.  Line no. 138-139 states that "A total of 232 samples were taken, collecting once or twice a week" however figure 2 shows that n = 232 only for the growing season of 2017 whereas n= 192 for GS 2018 and n= 128 for others. Kindly correct.

Response: Thank you for your precise comment. We deleted the sentence because it was inaccurate and superfluous.

8.  One-way ANOVA was performed to compare only soil temperature (line no. 144). As Figures S2 and S3 indicate that you performed ANOVA for soil moisture and air temperature also, correct your statement.

Response: Thank you for your precise comment. We revised this sentence as "One-way ANOVA was used to compare soil temperature, soil moisture and air temperature differences, respectively." See line no. 163, 164.

9.  General linear analysis was carried out between soil temperature and GHG fluxes only. The same analysis could be repeated for soil moisture also.

Response: Thank you for your precise comment. General linear analysis was carried out between soil moisture and GHG fluxes, Figure S6 was drawn and added to the Appendix. And the relevant sentences are added as "General linear analyses were

used to identify significant linear correlations and regressions between soil temperature and moisture variations and the responses of $Re$, $CH_4$ uptake, or $N_2O$ emissions, respectively." See line no. 164-170.

"However, $Re$, $CH_4$ uptake and $N_2O$ emission were no significant linearly correlated with soil moisture, respectively ($P \geq 0.05$; Figure S6)." See line no. 217, 218.

[Figure]

Figure S6 The relationship between ecosystem respiration ($Re$), $CH_4$ uptake and $N_2O$ emissions and soil moisture (at 10-cm depth) from October 2016 to September 2019. AW, warming throughout the year; NGW, warming in the nongrowing season only; GW, warming in the growing season only; NW, non-warming.

10. Use of variation partitioning analysis in figure 4 should be mentioned under methodology.

Response: Thank you for your precise comment. We added the sentence to our methodology, "variation-partitioning analysis was used to disentangled the influence of soil temperature and soil moisture on $Re$, $CH_4$ uptake, and $N_2O$ emission under the four treatments in the growing season and the non-growing season, respectively." See line no. 170-172.

**Results**

11. Results of all the GHG fluxes under warming have been given in terms of increase or decrease however, the ANOVA results do not show significant difference, which the control group (NW), the *Re* was decreased by 7.5% and 4.0% in the growing season and non-growing season, respectively, under AW" add "however non-significant" Alternatively, write line no. 175-179 (stating ANOVA results) before line no. 160, so as to report in the beginning only, that the differences were not significant.

Response: Thank you for your precise comment for the Results. We put the sentence of line no. 175-179 before line no. 160. See line no. 186-189.

12. Line no. 172: increase in $N_2O$ emission by 101.9% and 192.3% under AW and NGW in the growing season seems very high. Please check.

Response: Thank you for your precise comment. We checked and recalculated the data to make sure it was correct. Refer to the data in Table R1. To view the detailed data set, please visit the website: http://doi.org/10.5281/zenodo.4244207

Table R1 $N_2O$ emissions in NGS 2016-2017, NGS 2017-2018 and NGS 2018-2019. AW, warming throughout the year; NGW, warming in the nongrowing season only; GW, warming in the growing season only; NW, non-warming (Control group).

| | $N_2O$ flux ($\mu g$ N $m^{-2}$ $h^{-1}$) | | | |
| --- | --- | --- | --- | --- |
| | AW | NGW | GW | NW |
| NGS 2016-2017 | 1.12 | 1.70 | 0.51 | 0.50 |
| NGS 2017-2018 | 1.57 | 2.26 | 0.64 | 0.68 |
| NGS 2018-2019 | 1.68 | 2.37 | 0.49 | 0.99 |
| Mean | 1.46 | 2.11 | 0.55 | 0.72 |
| SD | 0.30 | 0.36 | 0.09 | 0.25 |
| Percentage | 101.9% (AW-NW)/NW | 192.3% (NGW-NW)/NW | | |

13. The authors may fit an exponential curve to determine the relationship between Re and soil temperature at 10 cm depth. Figure S5 a indicate towards an exponential pattern.

Response: Thank you for your precise comment. As your suggested, an exponential curve is more appropriate than the linear fitting to determine the relationship between *Re* and soil temperature at 10 cm depth. Through the value of the exponential function, we can well judge the response difference of *Re* with temperature increase under different warming treatments. We revised the Figure S5 a as an exponential pattern. And the related sentences in the manuscript were revised simultaneously. See line no. 164-170 "Nonlinear regression analyses (exponential growth, Single, 3 Parameter) was used to identify the relationship between ecosystem respiration (Re) and soil temperature (at 10-cm depth) from October 2016 to September 2019." See line no.223-225 "Under the four warming treatments, *Re* was significantly exponential growth

correlated with soil temperature ($P < 0.05$; Figure S5 a)."

[Figure]

Figure S5 The relationship between ecosystem respiration (*Re*), CH₄ uptake and N₂O emissions and soil temperature (at 10-cm depth) from October 2016 to September 2019. AW, warming throughout the year; NGW, warming in the nongrowing season only; GW, warming in the growing season only; NW, non-warming.

**Discussion**

14. Results should report response rates (only given in discussion section). The low $r^2$ value of linear regression in Figure 3 (where significant) should be discussed.

Response: Thank you for your precise comment. Response rates (RR) of GHG fluxes are mainly used to reveal the effect of temperature change on GHG flux. It is more valuable to clarify the relationship between temperature change and the GHG flux change in the Discussion. In response to the previous comment, we simply analyzed the microclimate variation in our experimental method, so we did not mention the RR value of seasonal asymmetric warming in our results.

The r$^2$ value of linear regression in Figure 3 was low, but the correlation was significant (Figure 3 a, b, c, and g). It is not appropriate to use the determination coefficient (r$^2$) in the Figure 3, so we revised the Figure 3 and used the value of correlation coefficient (r) to represent the correlational relationship between soil temperature change and GHG flux response rate (RR).

[Figure]

Figure 3 Response (presented by linear regression) of variation in ecosystem respiration (*Re*), CH$_4$ uptake, and N$_2$O emission to changes in soil temperature under AW, NGW and GW conditions in the alpine grassland, from 2016 to 2019. RR, the natural logarithm of the response ratio of the mean value of the chosen variable in the warming group to that in the control (NW) group. $\triangle$ST$_{AW}$, soil temperature of AW minus that of NW; $\triangle$ST$_{CW}$, soil temperature of NGW minus that of NW; $\triangle$ST$_{WW}$, soil temperature of GW minus that of NW; AW, warming throughout the year; NGW, warming in the non-growing season only; GW, warming in the growing season only; NW, non-warming.

15. Line no. 251-260 merely gives results of variation partitioning analysis without any

interpretation. This analysis should be mentioned in methodology and in result section.

Response: Thank you for your precise comment. This section (line no. 251-260) has been moved to the results, see line no. 197-208.

**Minor edits**

Line 1: Although it is known that ecosystem respiration means $CO_2$ emissions, why not specify $CO_2$ instead for coining "respiration" as done for other gases (as done in the previous draft). This will avoid ambiguity in title. for example, "Different responses of ecosystem $CO_2$ and $N_2O$ emissions and $CH_4$ uptake to seasonally asymmetric warming in an alpine grassland of the Tianshan Mountains" Also delete comma after $CH_4$ uptake.

Response: Thank you for your precise comment. We revised the Title as "Different responses of ecosystem $CO_2$ and $N_2O$ emissions and $CH_4$ uptake to seasonally asymmetric warming in an alpine grassland of the Tianshan Mountains" See line no. 1.

Line 16-19: specify percentage increase for each GHG flux.

Response: Thank you for your precise comment. We revised the sentence as "Furthermore, warming during the non-growing season increased $Re$ and $CH_4$ uptake by 7.9% and 10.6% in growing season, 10.5% and 9.2% in non-growing season, respectively. However, the increase in $N_2O$ emission in the growing season was mainly caused by the warming during the growing season (by 29.7%), the warming throughout the year and warming during the non-growing season increased $N_2O$ emissions by 101.9% and 192.3% in non-growing seasons, respectively." See line no. 17-22.

Line no. 26: remove comma after annual warming.

Response: Thank you for your precise comment. We removed comma after annual warming. See line no. 32.

Line no. 31: (i) Write greenhouse gas fluxes instead of flux, (ii) as the manuscript doesn't include temperature sensitivity as objective and it has not been calculated, it is not logical to use it as a keyword.

Response: Thank you for your precise comment. we revised the Keywords as "Alpine steppe; Extreme climatic event; Greenhouse gas fluxes; Warming of open-top chambers" See line no. 37.

Line no. 35-37: Shorten the sentence as "The global surface temperature increased by about 0.85°C from 1880 to 2012 and is expected to increase by about 1.1–6.4°C by the end of this century (IPCC, 2007, 2013).

Response: Thank you for your precise comment. We shortened the sentence as "The global surface temperature increased by about 0.85°C from 1880 to 2012 (IPCC, 2013). Furthermore, the temperature is expected to increase by about 1.1–6.4°C by the end of this century (IPCC, 2007, 2013)." See line no. 41, 42.

Line no. 38: remove comma after scale.

Response: Thank you for your precise comment. We removed the comma after scale. See line no. 44.

Line no. 52: The warming or the temperature sensitivity of the GFG fluxes have not been evaluated in the study and hence "and their sensitivity to warming" may be removed.

Response: Thank you for your precise comment. We removed "and their sensitivity to warming". See line no. 62.

Line no. 60. Remove space between numeric and percentage sign. Follow this in the entire manuscript.

Response: Thank you for your precise comment. We removed space between numeric and percentage sign in the entire manuscript.

Line no. 62: what do you mean by $CO_2$ fluxes? Is it respiration (if yes is it soil or ecosystem) or photosynthesis or both? Consider this in Line no. 78 also.

Response: Thank you for your precise comment. We revised the "$CO_2$ fluxes", "C fluxes" as "ecosystem respiration" See line no. 73, 74, 97, 98.

Line no. 63-67: as you are stating the result of specific study (Lin et al., 2015), it is better to start the sentence as "Lin et al. (2015) reported….". Also give the percentage increase in $CH_4$ uptake under growing season also. Alternatively, you may add more references of the studies showing similar results, in this the percentage increase may be removed.

Response: Thank you for your precise comment. We revised the sentence as "Lin et al. (2015) reported…." See line no. 86.

Lime no. 85: replace GHG flux with GHG fluxes. Follow this in Line no. 88, 98, 100.

Response: Thank you for your precise comment. We replaced "GHG flux" with "GHG

fluxes" in Line no. 88, 98, 100.

Line no. 111: add space between −4.8 and °C.

Response: Thank you for your precise comment. We added space between −4.8 and °C.

Line no. 157-159: Figure 1 does not show the annual mean values of each flux but the variations during each year and hence the reference to figure 1 in this sentence is redundant.

Response: Thank you for your precise comment. We deleted the reference to figure 1 in this sentence.

**Suggestions**

1. Microclimatic parameters such as air and soil temperature and soil moisture are important to understand variations in seasonal, inter-annual and the asymmetric warming effect on GHG fluxes. Hence these should be included in the main text and their methodology and results should be stated and used while interpreting warming or inter-annual effects on GHG fluxes.

Response: Thank you for your suggestion. In our manuscript, we have paid attention to the influence of soil temperature and humidity changes on greenhouse gas fluxes, and we will analyze these good suggestions you mentioned in a more delicate way in the later research.

2. Calculate $Q_{10}$ values (at least for ecosystem respiration). This will give you a more direct indication of temperature sensitivity changes with warming. As most studies use this approach, it will be useful for comparison.

Response: Thank you for your precise comment. Taking into account methane absorption and nitrous oxide emissions, we used RR instead of Q10. Your suggestion is very good. $Q_{10}$ is calculated in our future study on the sensitivity of ecosystem respiration to temperature caused by warming.

3. In Figure 2, it is suggested to add boxes for mean (entire study period) of each flux rates during growing and non-growing season under four treatments along with ANOVA results (as letters).

Response: Thank you for your precise comment. We revised the Figure 2 and add boxes for mean (red line) under four treatments along with ANOVA results (as letters).

[Figure]

Figure 2 Boxplot presentation of variations in ecosystem respiration (*Re*), $CH_4$ uptake, and $N_2O$ emission under four treatments in the growing season and non-growing season from October 2016 to September 2019. The median and mean are represented by the black and red lines in the box, respectively. The box (the interquartile range) represents the middle 50% of the data, whereas the whiskers represent the ranges for the bottom 25% and the top 25% of the data values, excluding outliers. GS, growing season; NGS, non-growing season; AW, warming throughout the year; NGW, warming in the non-growing season only; GW, warming in the growing season only; NW, non-warming. No significant differences among AW, NGW, GW, and NW were reported from ANOVA; data points are the mean ± standard error. One-way ANOVA results of *Re*, $CH_4$ uptake and $N_2O$ emissions among the four warming treatments were not significant, except that the $CH_4$ uptake in the GS 2019 under the GW treatment was

significantly higher than that of AW and NGW treatment ($P < 0.05$).

---

## Author Response (AR3)

**A point-by-point reply**

Dear Associate Editor and Anonymous Referee #3:

Thank you for your comments and the reviewers' comments concerning our manuscript entitled "Different responses of ecosystem $CO_2$ and $N_2O$ emissions and $CH_4$ uptake to seasonally asymmetric warming in an alpine grassland of the Tianshan Mountains" (MS No.: bg-2020-396). Those comments are all valuable and very helpful for revising and improving our paper, as well as the important guiding significance to our researches. We have studied those comments carefully and have made a point to point reply and correction. Revised portion are marked in red color in this manuscript. Specific corrections and responds to the Referee's comments are listed as follows:

Associate Editor Decision: Publish subject to technical corrections (28 Apr 2021) by Ben Bond-Lamberty

Comments to the Author:

This manuscript was re-read by the reviewers, and both find it significantly improved. I agree—thanks for your careful revisions, which have dramatically improved the clarity of the text and strengthened the overall study.

There do remain a few changes that could be made to further clarify and improve the text; these are all optional, but I strongly recommend them. See short list below. Regardless, I am pleased to recommend this ms for acceptance.

Minor comments

1. Line 14: Be concise, i.e. "The annual mean…" (no need for "Our results indicated that")

**Response:** Thank you for your precise comment. We revised as "The annual mean…", see Line 14 in the revised manuscript.

2. L. 101-104: for clarity, would be good to start new paragraph with "In this study, we… [describe in a single sentence]." Then "we hypothesized that Re, CH$_4$ update, and N$_2$O emissions would be simulated by warming, but gradually attenuated…"

**Response:** Thank you for your precise comment. We revised as "In this study, we hypothesize the stimulatory effect of warming…". See Line 102.

3. L. 104-107: could remove

**Response:** Thank you for your precise comment. We removed this sentence.

4. L. 187: start new paragraph

**Response:** Thank you for your comment. We revised this paragraph.

5. L. 206: "Re exhibited exponential growth"

**Response:** Thank you for your comment. We revised this sentence. See Line 212.

6. L. 212: "were not"

**Response:** Thank you for your comment. We revised as "…were not…". See Line 217.

7. L. 216: see #1 above, same comment here (also in line 267)

**Response:** Thank you for your comment. We revised the sentences. See Line 221, 273 and 275.

8. L. 241: "In this study, warming increased…"

**Response:** Thank you for your comment. We revised as "In this study, warming increased…" See Line 248,249.

9. L. 292: what about availability of analysis code? Please include in Zenodo!

**Response:** Thank you for your precise comment. All statistical analyses were conducted using SPSS (version 20.0) (IBM, Armonk, NY, USA) with the statistically significant difference threshold set at $P < 0.05$. Our statistical analysis of the data did

not use any code.

**Report #1 from Anonymous Referee #3**

The manuscript titled "Different responses of ecosystem $CO_2$ and $N_2O$ emissions and $CH_4$ uptake to seasonally asymmetric warming in an alpine grassland of the Tianshan Mountains"

by Gong et al. may be accepted after the following minor edits:

1. The authors, in this revision, state that they measured humidity (line no. 136). The variations in humidity (I assume relative humidity) under warming as given for other parameters should be provided in supplementary. Alternatively, I suggest that humidity may be removed as it is not discussed further in the manuscript.

**Response:** Thank you for your precise comment. We removed "…and humidity…". See Line 140 in the revised manuscript.

2. In line no. 138 correct "OCTs" to "OTCs"

**Response:** Thank you for your comment. We revised it. See Line 142.

3. In figure 3, If you provide correlation (r value) instead of regression (r2 value) then in line no. 476 (figure 3) correct "presented by linear regression" to "presented by linear correlation".

**Response:** Thank you for your precise comment. We revised this sentence. See Line 470-474.

4. My earlier suggestion regarding figure 2 was to add additional boxes to represent overall variations in GHG fluxes (entire sampling period from 2016-2019). It will read as (in X-axis): GS 2017 GS 2018 GS 2019 GS Mean Do the same for NGS. Mean lines (shown in red) are not required.

**Response:** Thank you for your precise comment. We removed the Mean lines (shown in red) in the Figure 2.